# Massively parallel quantification of mutational impact on IAPP amyloid formation

Marta Badia [1,2], Cristina Batlle[1] & Benedetta Bolognesi [1] ✉

Amyloid fibrils formed by the islet amyloid polypeptide cause pancreatic beta-cell damage, resulting in reduced insulin secretion and type 2 diabetes. Changes in the amino acid sequence of this peptide can influence its aggregation rate, and animals expressing variants that do not form amyloids do not develop type 2 diabetes. Conversely, specific single amino acid changes can accelerate the aggregation rate of this peptide. Here, we employ deep mutational scanning to measure the ability of 1916 islet amyloid polypeptide variants, including substitutions, insertions, truncations and deletions, to nucleate amyloids. Our results identify a continuous stretch of residues from 15 to 32 that is particularly sensitive to mutation. This region, which is likely structured in amyloids, matches the core of the early aggregated species formed by this peptide in vitro. Within this region, mutations in residues 21 to 27 have a substantial effect, suggesting tighter structural constraints. Finally, we compare the mutational atlas of the islet amyloid polypeptide to that of amyloid beta - the peptide that aggregates in Alzheimer's disease - and find that mutations that slow down nucleation correlate between the two amyloids, but mutations that accelerate nucleation in one amyloid cannot be used to predict mutational effects in the other.

Amyloids formed by the islet amyloid polypeptide hormone (IAPP) are among the first ever amyloids observed and extracted from human tissue, more than 120 years ago[1], decades before evidence of amyloids was observed in dementia brains.

It is now established that IAPP amyloids cause pancreatic beta-cell damage, leading to a decline in insulin secretion and type 2 diabetes (T2D)[2], a debilitating condition that affects over 100 million patients worldwide[3]. While IAPP is highly conserved among mammals, IAPP variants that differ in less than a handful of positions from the human sequence don't form amyloids, and animals expressing these sequences, such as bear or mouse, do not spontaneously develop T2D[4,5]. It is, however, enough to express human IAPP in a mouse model to cause the formation of pancreatic amyloids and for diabetes to occur[6,7]. Pramlintide, an IAPP analog that does not form amyloids, is currently approved as a T2D treatment, with a need for the design of alternative IAPP analogs with improved solubility[8].

Early aggregates formed by IAPP have been shown to be more damaging for pancreatic beta-cells[2,9–11], highlighting the need to understand and control amyloid nucleation, the initial event and rate-limiting step of the amyloid reaction[12]. The challenges of quantifying amyloid nucleation rates for multiple variants in the same conditions have so far partially limited our understanding of the sequence-nucleation relationship for IAPP[13]. Only a few IAPP variants have been characterized, and the vast majority have been shown to decrease IAPP nucleation, making it difficult to predict whether novel population variants might increase the risk of developing T2D.

Several studies have proposed an analogy between IAPP and another extracellular peptide, amyloid beta (Aβ42), the main

[1]Institute for Bioengineering of Catalonia (IBEC), The Barcelona Institute of Science and Technology, Barcelona, Spain. [2]Universitat de Barcelona, Barcelona, Spain. ✉e-mail: bbolognesi@ibecbarcelona.eu

component of the plaques that deposit in Alzheimer's disease (AD) brains[14]. Sequence alignment of the two proteins shows 56% sequence similarity, with 9 positions (24%) having identical amino acids. Four of these identical amino acids are located in the previously described core of the fibrils[14–18] formed in vitro by both proteins (NxGAI, positions 22, 24–26 in IAPP and 27, 29–31 in Aβ42). Different structural polymorphs have been described for fibrils formed by the two peptides in vitro or extracted from human tissue (8 for IAPP[14,19–23] and 10 for Aβ42[24–29]). Among these, two specific Cryo-EM structures of in vitro fibrils display striking structural alignment[14,20]. The two peptides have also been reported to interact and co-aggregate in vitro and in yeast cells[30–33].

We reasoned that a systematic approach to probe the similarities between these two sequences that form amyloids in physiological conditions is the massively parallel quantification of mutational impact to understand whether mutations speed up or slow down the amyloid reaction in a similar way.

To achieve this, we employed a multiplexed assay of variant effect (MAVE), an approach that combines systematic mutagenesis with quantitative phenotypic measurements. MAVEs generate detailed sequence-function maps that reveal the effects of each variant in parallel[34–37]. By linking variant libraries to high-throughput sequencing readouts, these assays scale to quantify tens of thousands of mutations in parallel, allowing highly resolved and comprehensive maps of sequence-function relationships. Such approaches have been applied to a wide range of proteins and molecular phenotypes, providing systematic insights into sequence-function relationships[38,39].

We have previously mapped the complete mutational landscape of the Aβ42 peptide[40] and have gathered insights into pathogenic gain of function (GOF) from systematically characterizing insertions and deletions (indels)[41]. Here, we employ a similar strategy to quantify the impact of substitutions, insertions, and deletions in IAPP, for a total of 1916 variants. Besides comprehensively mapping the IAPP mutational landscape, we identify a likely structured core of the nucleating IAPP fibrils that is consistent with and adds granularity to what has been suggested by previous studies[14–18,42]. The resulting dataset reveals that GOF, i.e., the acceleration of amyloid nucleation, can result from all different classes of mutations, highlighting the need to improve our ability to measure, understand, and predict the impact of indels in human sequences.

We also systematically compare the outcome of mutations in IAPP and Aβ42 and find significant similarities between mutational effects in IAPP and Aβ42 when it comes to mutations that reduce amyloid nucleation, suggesting a common mechanism of disruption of amyloid nucleation. However, the effects of GOF mutations, which instead speed up nucleation and are more frequent outside the core amyloid region of both peptides, are distinct in the two datasets. Overall, one mutational dataset cannot be used to predict the effect of mutations in the other. This highlights the challenge of predicting pathogenic GOF from primary sequence and the need to produce similar quantitative datasets for all amyloids observed in human disease.

## Results

### Massively parallel quantification of IAPP amyloid nucleation

We synthesized a library encompassing six classes of mutations: single amino acid substitutions, insertions, and deletions in IAPP, as well as truncations, larger internal deletions, and designed sequence variants simulating polymerase slippage (i.e., replication errors that create small insertions or deletions due to transient misalignment of the polymerase with the template). We then quantified the ability of each sequence to form amyloids using a cellular assay where IAPP variants are fused to the nucleation domain of the yeast prion Sup35 (Sup35N), since independent expression of Sup35N and IAPP variants does not result in recruitment of endogenous Sup35p, as reported by the Chernoff laboratory[43] and confirmed here (Supplementary Fig. 1a).

Nucleation of endogenous Sup35p is required for yeast survival in the lack of adenine, so that fitness-based selection becomes possible and that variants that nucleate amyloid fibrils inside yeast cells get enriched upon selection (Fig. 1a). The relative enrichment of each variant is calculated from sequencing the variant library before and after selection, providing an estimate of the ability of each variant to nucleate amyloids (see the "Methods" section). Testing of individual Sup35N-IAPP variants spanning a range of nucleation scores confirmed that differences in nucleation were not attributable to cellular toxicity, as cells expressing distinct variants featured comparable yeast growth upon protein expression, irrespective of their nucleation scores (one-way ANOVA; Supplementary Fig. 1b).

As a result, we obtained a dataset reporting nucleation scores and associated error terms for 1916 IAPP variants that belong to all classes of mutations (Fig. 1g). Nucleation scores are reproducible across biological replicates (Supplementary Fig. 1c, d) and they also correlate well with measurements of individual variants' growth in selective conditions (Fig. 1e). By comparing nucleation scores to the results of over ten previous studies[13,44–57] where the kinetics of amyloid formation of IAPP variants ($n = 38$) were followed in vitro by Thioflavin-T fluorescence, we show that nucleation scores capture the direction of the mutational effects in speeding up or slowing down the rate of amyloid formation (Fig. 1c, Supplementary Data 1). Our assay also captures the different amyloid propensity of IAPP animal variants that have previously been characterized[4,46,58] to form or not to form aggregates in vitro and in vivo (Fig. 1d). Importantly, nucleation scores are not explained by simple changes in expression levels (Supplementary Fig. 1e), and they scale with the half-time of the aggregation reaction when synthetic IAPP variants, which are not fused to Sup35N, aggregate in the presence of Thioflavin-T (Supplementary Fig. 1f). Overall, thanks to this deep mutational scanning approach we obtain quantitative estimates for all IAPP variants in one unique experimental set-up, overcoming some of the challenges involved in quantifying in vitro rates for more than a handful of variants in identical conditions. For comparison, at most four variants were studied in parallel in previous studies, and quantitative estimates of the rate constant were only obtained for IAPP wild type (WT) and p.Ser20Gly[13].

### Distribution of mutational effects across classes of mutations

The distribution of nucleation scores (NS) in each class of mutations reveals that mutations in IAPP are overall more likely to decrease the ability of the peptide to nucleate amyloids, with 64.6% of mutations slowing down aggregation (NS−), while 15.3% of mutations behave similarly to WT IAPP and only 20.1% speed up its aggregation (NS+). In those classes of mutations, where changes involve the removal of more than one amino acid such as multiple amino acid deletions and truncations from the N-terminus and the C-terminus, an even larger percentage of variants reduce nucleation (Fig. 2a, 88.3% of NS− multiple amino acid deletions, 100% of NS− N-terminal truncations, and 97% of NS− C-terminal truncations).

### Single amino acid substitutions highlight a structured core between residues 15 and 32

Substitutions to proline and glycine (Fig. 2d) reduce (or maintain) nucleation in a continuous stretch from residue 15 to 32, with the only exception of p.Ser20Gly, suggesting this region is likely to be structured in the amyloid fibrils nucleated by IAPP and where side-chain constraints are tighter. This region matches the structured region in all fibrils formed by IAPP in vitro, as well as in those seeded by samples extracted from T2D patients. Within this region, we find the lowest nucleation scores in the NNFGAIL stretch (Fig. 2b, c), which has been shown to be essential for IAPP aggregation in vitro and one of the shortest peptides known to form amyloids in vitro[14,59,60]. Mutations of Ala 25 represent the only exception in this window, in line with in vitro evidence demonstrating that substitutions of Ala 25 can maintain

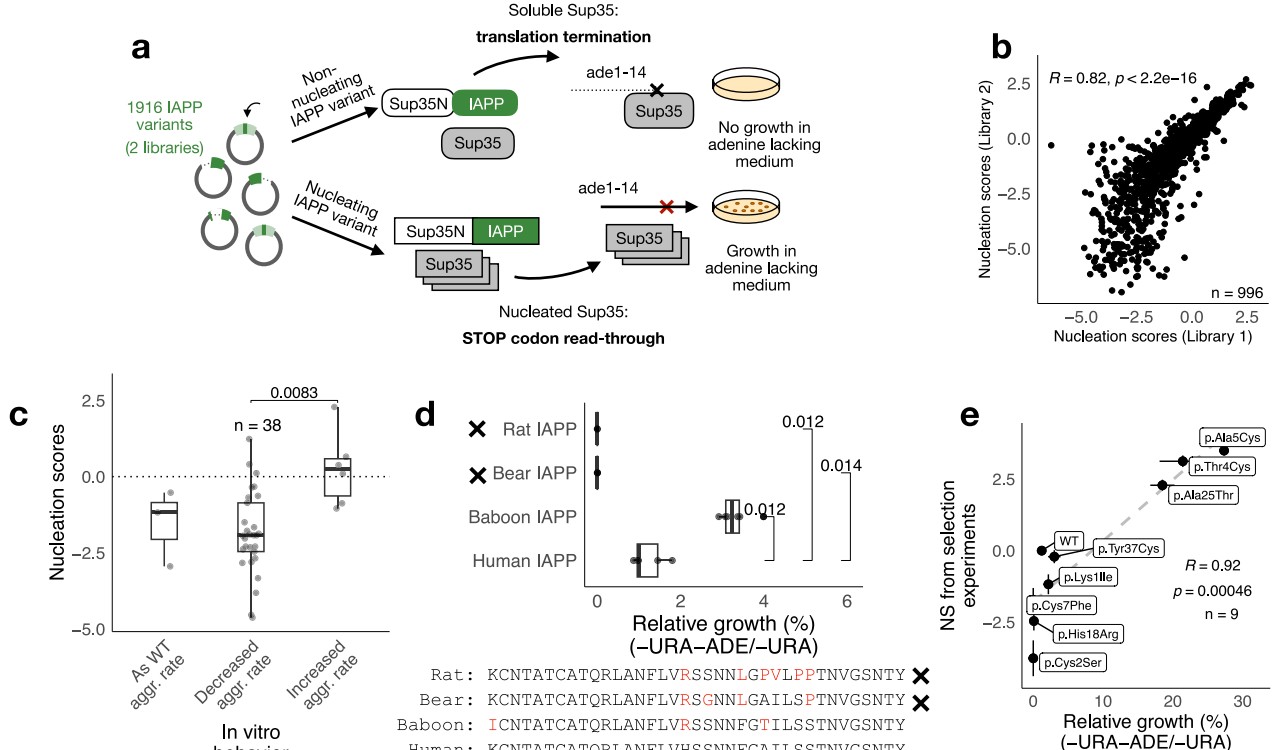

**Fig. 1 | Deep mutagenesis of IAPP. a** Description of the IAPP library and schematics of the in vivo selection experiment. IAPP variants are fused to Sup35N. In a strain with a premature STOP codon in the *ade1* reporting gene, Sup35N-IAPP fusions that seed aggregation of full-length Sup35p will cause STOP codon read-through and confer cells the ability to grow in adenine-lacking medium. **b** Correlation of nucleation scores for variants shared between the 2 libraries used in this study. Nucleation scores of the common variants were fitted using a Deming regression, and the resulting regression equations were used to normalize scores across libraries for combined analysis. The number of shared variants between libraries, Pearson correlation coefficient (*R*, two-sided), and *p*-values are indicated on the plot. **c** Nucleation scores of 38 IAPP variants grouped by their reported ability to form aggregates in vitro. For each study, variants were compared to the wild-type IAPP reported in that same study (see Supplementary Data 1) and categorized accordingly on the x-axis. Boxes indicate the median and first (Q1) and third (Q3) quartiles; whiskers extend to the smallest and largest values within 1.5 × IQR of the lower and upper quartiles, respectively. Statistical differences between categories were assessed using a two-sided Wilcoxon rank-sum test with Benjamini–Hochberg correction for multiple comparisons; significant differences are indicated in the plot with exact p-values. **d** Relative growth of cells expressing IAPP sequences from different mammals quantified as percentage of colonies grown in selective conditions (-ADE -URA) over colonies grown in non-selective conditions (-URA) (*n* = 4, with 5 biological replicates each). Boxes indicate the median and first (Q1) and third (Q3) quartiles; whiskers extend to the smallest and largest values within 1.5 × IQR of the lower and upper quartiles, respectively. Statistical differences between human IAPP and animal IAPP variants were assessed using a two-sided Wilcoxon rank-sum test with Benjamini–Hochberg correction for multiple comparisons; significant differences are indicated in the plot with exact p-values. Variants reported in the literature as non-nucleating are marked with a cross (×) next to their y-axis labels and in the amino acid sequence written below the plot. Amino acids different from the human IAPP sequence are colored in red. **e** Correlation between nucleation scores from selection experiments and relative growth measured for individual variants in the absence of competition, quantified as percentage of colonies grown in selective conditions (-ADE -URA) over colonies grown in non-selective conditions (-URA) (*n* = 9, with 3 biological replicates each). Error bars indicate the standard deviation of the mean estimations. Correlation coefficients (*R*) and *p*-values were calculated using a two-sided Pearson correlation test.

nucleation, including the p.Ala25Pro variant, which alone or in combination with p.Ser29Pro still forms amyloid fibrils[58]. The impact of mutations in the NNFGAIL stretch appears to be dominant: double mutants containing the p.Ala25Thr mutation (NS+) feature increased aggregation compared to the other single mutant, while the opposite happens in double mutants containing p.Phe23Leu (NS−), where amyloid formation is reduced relative to the other single substitutions (Fig. 2e).

We also find that mutating Cys residues at positions 2 and 7 drastically reduces nucleation and that instead mutations to Cys at the first 9 positions speed up the rate of amyloid nucleation, with p.Ala5-Cys, p.Tyr9Cys, and p.Tyr4Cys scoring as the strongest nucleators among all IAPP substitutions (Fig. 2b).

While this fusion peptide may not recapitulate all of the modifications that in vivo characterize biologically active IAPP, such as C-terminal amidation and formation of the disulfide bond between Cys 2 and Cys 7, the mutational landscape of IAPP substitutions presented

here identifies a likely structured region (15–32) in IAPP amyloids as well as an inner stretch of core residues (21–27) that are essential for amyloid nucleation, both of which are in line with previous results obtained in vitro with unfused oxidized peptide. Nucleation scores also scale with the half-time of the in vitro aggregation reaction for unfused IAPP variants (Supplementary Fig. 1f), overall suggesting the Sup35N-IAPP fusion employed in the selection assay can accurately capture the essence of IAPP nucleation.

## Mutational impact is consistent with an S-fold subunit arrangement

Four structures of fibrils formed in vitro by recombinant or synthetic IAPP have been resolved by Cryo-EM[14,19,20]. In all of these, the first 11 or 12 N-terminal residues are unresolved, suggesting that, regardless of the state of the disulfide, this region is either disordered or displays structure heterogeneity in the mature fibril arrangement[5]. In another set of four fibril structures, seeded by patient extracts, only residues

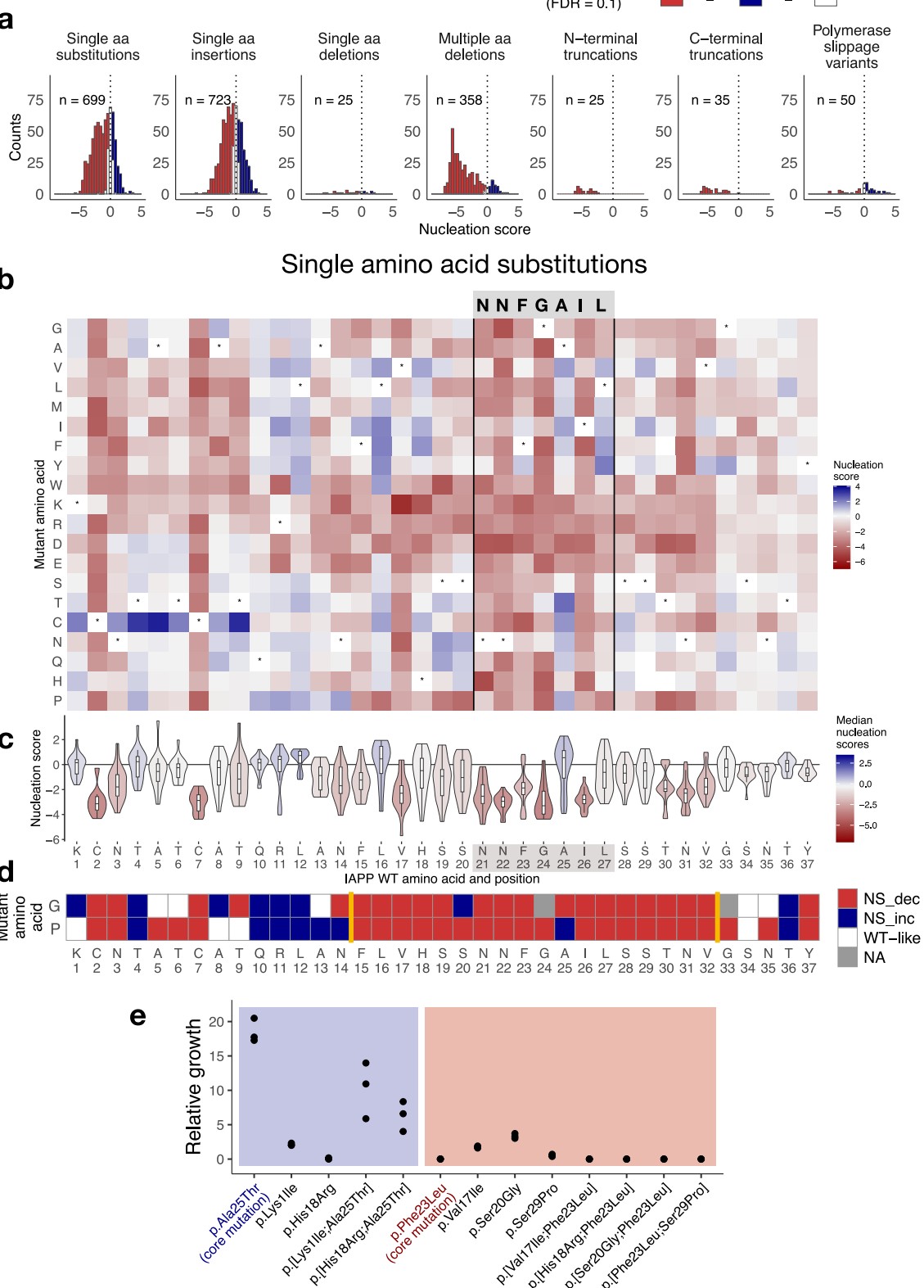

**b**

**Single amino acid substitutions**

**e**

1–5 are unresolved, but Cys 7 is not engaging in a disulfide[21]. Overall, we find 61 GOF substitutions of residues 1–12 that speed up amyloid formation of IAPP (Supplementary Fig. 2a, b).

We evaluated the match between mutational impact and the structural arrangement of IAPP fibrils by correlating average nucleation score per residue to the extent to which side chains are buried in each IAPP fibril structure (accessible surface area, ASA). Based on a

previous mutational scan of Aβ42[41], the expectation is that – if our assay is tracking the aggregation of IAPP into structures that are similar to those reported so far – then mutating those side chains buried in the core of amyloid fibrils should slow down nucleation the most. We find a stretch of residues (aa 21–27, with the exception of Ala 25) where more than 75% of single amino acid substitutions per position decrease the nucleation, prioritizing residues 21–27 as those more likely to get

**Fig. 2 | Mutational effects of IAPP amino acid substitutions. a** Distribution of nucleation scores and the number of variants grouped per mutation type. Bars are colored by the effect of mutations: increasing, decreasing, or having no effect on IAPP nucleation (FDR = 0.1). **b** Heatmap of nucleation scores of amino acid substitutions. The x-axis indicates the IAPP WT sequence, and the y-axis indicates the mutant amino acid. Variants not present in the library are represented in gray. Synonymous substitutions are indicated with "*". **c** Distribution of the nucleation scores per position. Violin plots summarize the effect of all single amino acid substitutions at each of the 37 IAPP positions. Violin plots are colored by the mean nucleation score per position. Boxes indicate the median and first (Q1) and third

(Q3) quartiles; whiskers extend to the smallest and largest values within 1.5 × IQR of the lower and upper quartiles, respectively. **d** FDR categories of nucleation scores for substitutions to proline and glycine along the IAPP sequence (FDR = 0.1). Missing substitutions are colored in gray. Yellow lines indicate a continuous stretch of 18 amino acids where mutations to proline and glycine (or both) decrease nucleation. **e** Relative growth of double mutants and corresponding single mutants, quantified as percentage of colonies grown in selective conditions (-ADE -URA) over colonies grown in non-selective conditions (-URA). At least 3 biological replicates per variant were performed.

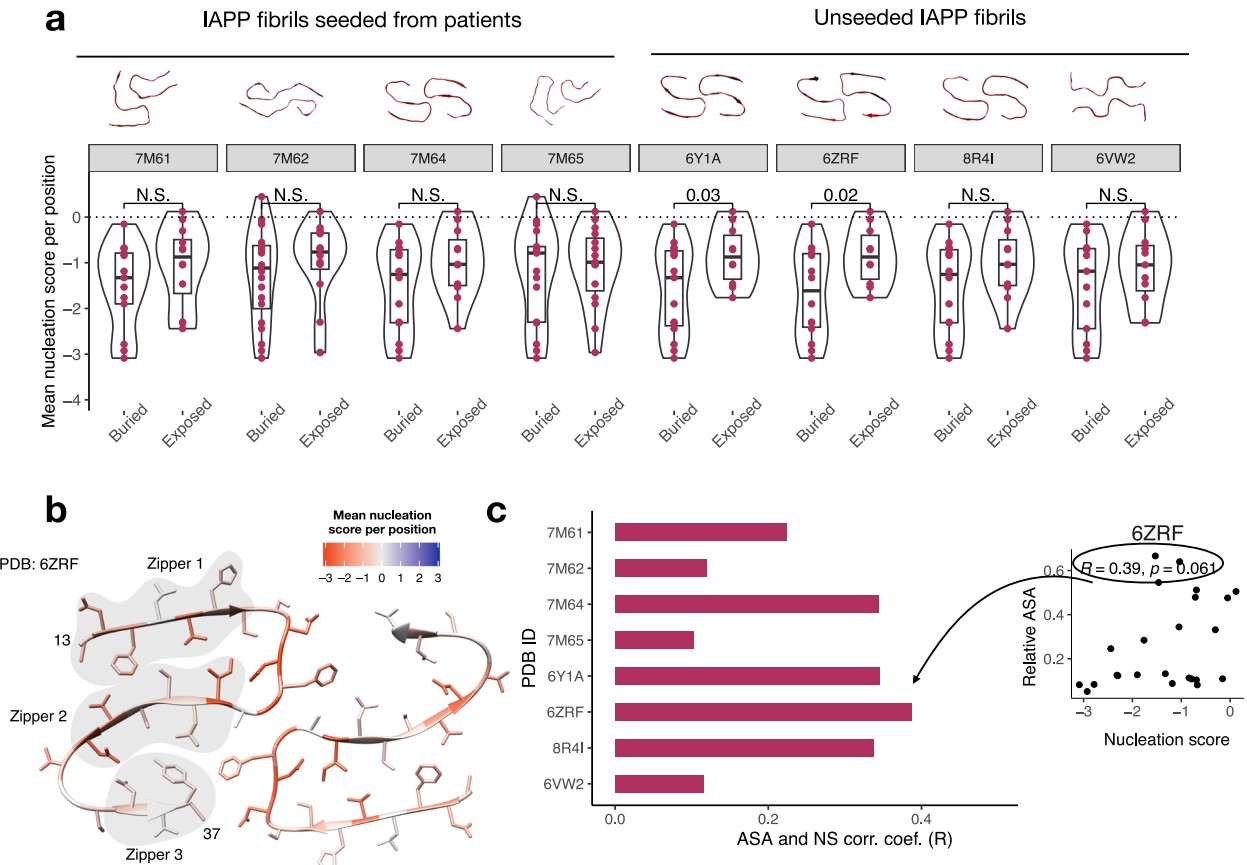

**Fig. 3 | Mapping mutational impact on IAPP amyloid structures. a** Distribution of the averaged nucleation scores per position of single amino acid substitutions of IAPP positions classified by their exposure in all the available IAPP fibrils PDB structures. Each point represents the mean nucleation score of a single residue, with the total number of points equal to the number of residues resolved in each structure (PDB 7M61[21], 7M64[21], 6Y1A[14], 8R4I[19] n = 25; PDB 7M62[21] and 7M65[21], n = 32; PDB 6ZRF[23] and 6VW2[20], n = 24). Boxes indicate the median and first (Q1) and third (Q3) quartiles; whiskers extend to the smallest and largest values within 1.5 × IQR of the lower and upper quartiles, respectively. Residues with ASA > 0.25 are

considered exposed. Differences between buried and exposed residues were tested using unpaired Welch's two-sample t-tests. Significant p-values are noted. **b** IAPP structure (PDB: 6ZRF[23]) colored by the median mutational effect of amino acid substitutions per position. **c** Correlation coefficients (R) of the correlations of mean nucleation scores per position and available surface area extracted from all available PDB structures (left). Representative scatter plot illustrating the correlation (right). Correlation coefficients (R) and p-values were calculated using a two-sided Pearson correlation test.

structured in the initial events of IAPP aggregation (Supplementary Fig. 3). These correspond to residues forming the minimal IAPP fragment required for amyloid formation, NNFGAIL[14,61], and these positions also feature low relative ASA across all available IAPP fibril structures (Supplementary Fig. 3), as their side chains are buried in the fibril core. We note that mutations that affect the first of the two zippers of the S-fold structures have more drastic effects than those affecting the third (Fig. 3b).

We also find that average nucleation scores per position are significantly different between buried and exposed residues (Fig. 3a), and that the overall correlation between nucleation score and accessible surface area (ASA) is higher for those structures where each fibril

subunit adopts a S-fold (PDB: 6Y1A[14], 6ZRF[23], 8R4I[19], 7M64[21]) and lower for those that consist of different folds (Fig. 3c). We note that the S-fold is also the fold of the first species identified in the early time points of the time-resolved Cryo-EM characterization of the amyloid reaction for WT IAPP[23] and the IAPP p.Ser20Gly[22] variant (Supplementary Fig. 4, Supplementary Fig. 5).

## Half of the possible single amino acid insertions maintain or accelerate IAPP amyloid nucleation

We next evaluated the amyloid nucleation scores resulting from amino acid insertions, thus measuring the result of sequence alterations that extend the main chain. The insertions of prolines and glycines

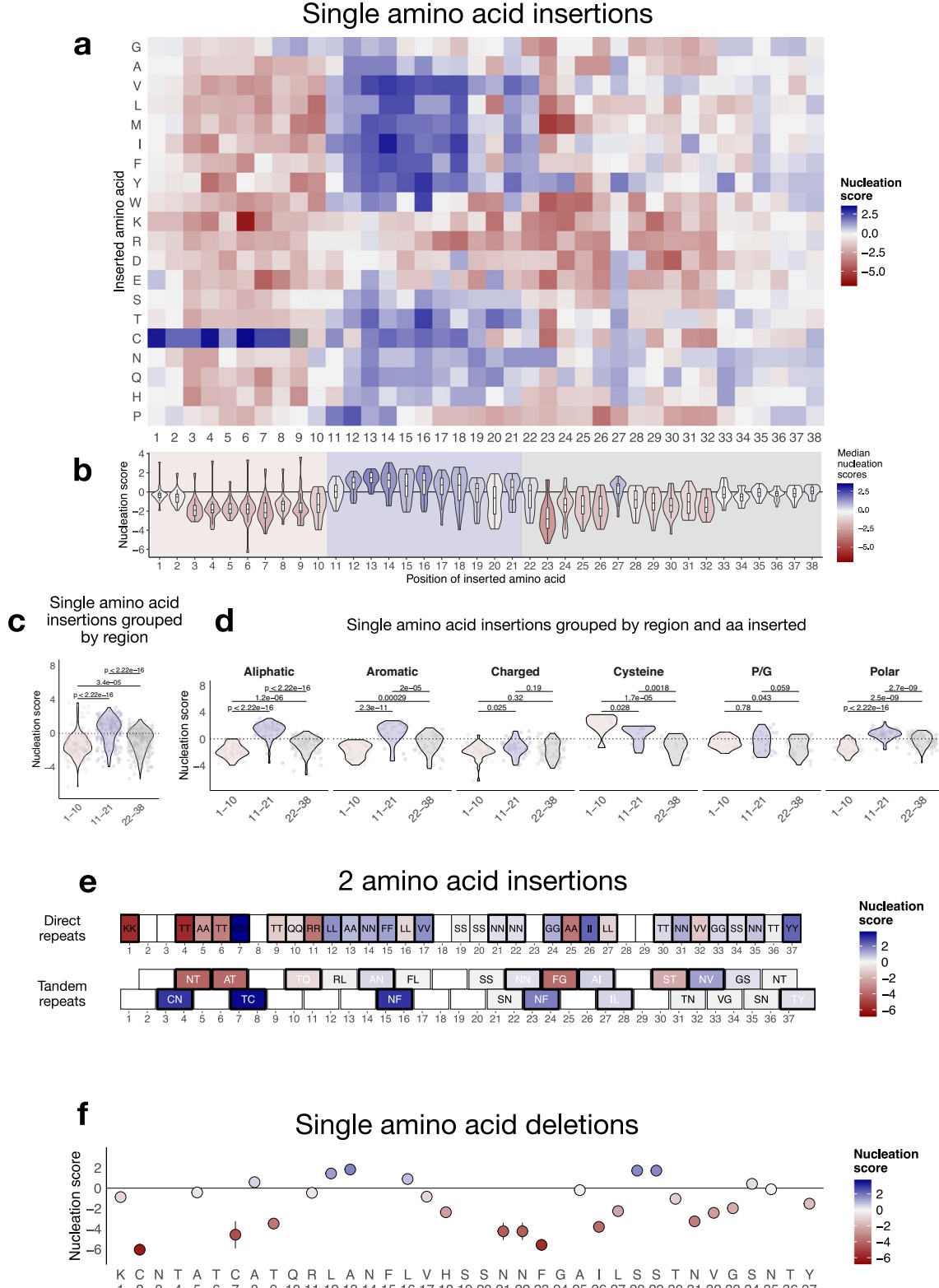

**a** Single amino acid insertions

**b**

**c** Single amino acid insertions grouped by region

**d** Single amino acid insertions grouped by region and aa inserted

**e** 2 amino acid insertions

**f** Single amino acid deletions

decrease nucleation in the same continuous stretch we identified by measuring substitutions (15–32) (Supplementary Fig. 6a). Within this window, all insertions after position 22, 23, 24, and 25 virtually reduce nucleation, confirming the coordinates of the essential inner core of IAPP amyloids. The N-terminus (aa 1–10) of the peptide is also sensitive to insertions (Fig. 4c). Here, the vast majority of these mutations reduce nucleation – with the exception of insertions of cysteines,

which, similar to substitutions to cysteines in this region, increase the rate of nucleation (Fig. 4d).

Systematic mapping of single amino acid insertions of nucleation scores per position (Supplementary Fig. 6a, b) also reveals a central region of the peptide, residues 11–22, where all insertions of aliphatic, aromatic, and polar residues increase nucleation (Fig. 4c, d). What is more, between residues 10 and 14, even the insertion of prolines and

**Fig. 4 | Mutational effects of IAPP amino acid insertions and deletions.**
**a** Heatmap of nucleation scores for single amino acid insertions. The x-axis indicates the position of the inserted amino acid, and the y-axis indicates the amino acid inserted. Variants not present are shown in gray. **b** Distribution of nucleation scores per position. Violin plots summarize the effect of all single amino acid insertions at each IAPP position. Violin plots are colored by the mean nucleation score per position. Boxes indicate the median and first (Q1) and third (Q3) quartiles; whiskers extend to the smallest and largest values within 1.5 × IQR of the lower and upper quartiles, respectively. Background colors indicate regions defined by mutational patterns (Supplementary Fig. 6b). **c** Distribution of nucleation scores for all single amino acid insertions, grouped by regions defined by mutational patterns. **d** Same as (**c**), but nucleation scores are grouped by the type of amino acid inserted. Pairwise comparisons were performed in (**c**) and (**d**) using two-sided

Student's *t*-tests with Bonferroni correction for multiple comparisons. **e** Nucleation scores of two-amino acid insertions resulting from polymerase slippage. The x-axis indicates the position after which each amino acid pair is inserted. The inserted amino acid pair is labeled within each tile, and tile colors represent nucleation scores. Variants with nucleation scores significantly different from WT (FDR = 0.1) are indicated with a wider black square. **f** Effect of single amino acid deletions on IAPP nucleation. The x-axis indicates the position of the deleted amino acid, and the y-axis shows the corresponding nucleation score and point color. Variants with nucleation scores significantly different from WT (FDR = 0.1) are indicated with a black circle. Vertical bars represent represent 95% confidence interval for the nucleation score estimates. The horizontal line indicates the nucleation score of WT IAPP.

glycines speeds up nucleation (Supplementary Fig. 6a), suggesting the presence of an element of secondary structure able to stabilize the IAPP monomeric ensemble against amyloid nucleation. When insertions disrupt it, nucleation becomes more likely, suggesting this region could represent a potential target for small-molecule binders able to modulate IAPP nucleation. The binding of chaperones stabilizing this region has been shown to inhibit fibril formation[5].

We have also evaluated the effect of double amino acid insertions that can result from polymerase slippage, i.e., the duplication of two-amino acids of the WT sequence, or of the same amino acid, repeated twice. Overall, these insertions of two amino acids have mixed effects, with variants that increase or decrease nucleation across the whole sequence (Fig. 4e, Supplementary Fig. 7). The effect of single amino acid insertions can be used to predict the impact of double amino acid insertions at the same position, when the same amino acid is repeated (R = 0.8, *p*-value = 3.6e-07) (Supplementary Fig. 8a), similar to what was found for the effect of insertions on protein folding[62]. Effects of single insertions can also be partially predicted on the basis of the single amino acid substitutions (R = 0.44, *p*-value = <2.2e-16) (Supplementary Fig. 9a, d), especially at positions within the central 11-21 region identified above (Supplementary Fig. 9b, c, e, f).

Finally, the insertion of just one single cysteine (due to single or double amino acid insertions) is enough to increase nucleation (Supplementary Fig. 10a). Together with the observation that nucleation is also increased for sequences where mutations result in two adjacent cysteines (Supplementary Fig. 10c), which could by no means form a disulfide bond, this also points at a distinct mechanism from cysteine oxidation by which increasing the number of cysteines just in the first 10 residues of the peptide can favor IAPP aggregation.

## 14 single amino acid deletions maintain or increase nucleation rates

In parallel, we measured the impact of single and multi amino acid deletions of the IAPP sequence (Fig. 4f, Supplementary Fig. 11). Their effects are more drastic when deletions cause loss of Cys 2, Cys 7, Thr 9, and of residues in the NNFGAIL stretch, with the exception of Ala 25, which, in line with the results of substitutions, seems dispensable for the formation of the inner core of IAPP amyloids. Deletions of single residues from position 11 to 16 increase or maintain nucleation, suggesting that alterations of the main chain in this region, other than insertions, can unlock faster nucleation. Finally, the individual loss of Ser28 or Ser29, which results in two identical sequences, increases nucleation.

Most of the large multi amino acid deletions disrupt nucleation, but 22 deletions of just 2–4 amino acids, at different positions along the sequence, can increase it (Supplementary Fig. 11). Among these, we highlight those deletions that don't reduce the number of cysteines, but rather bring the WT cysteines closer to each other, significantly speeding up nucleation (Supplementary Fig. 10b). We also observe a contrast between the effect of the loss of two serines, just before or just after the inner core NNFGAIL (S19, S20). The first double deletion

(Δ19-20) speeds up the aggregation process, while the latter (Δ28-29) slows it down, in contrast with the individual loss of Ser28 and Ser29. Finally, deletion of the entire 10–14 stretch (QRLAN) and other shorter deletions in this region result in sequences that nucleate faster than WT, further suggesting the presence of a gate-keeping element of secondary structure in this region of IAPP, which in the WT sequence limits aggregation rates.

## The same types of mutations decrease nucleation in IAPP and amyloid beta

To assess the similarity of mutational impact in different amyloidogenic peptides, we compared the IAPP nucleation landscape with that of Aβ42[41]. We find a positive correlation between the average effect of substitutions to (R = 0.75, *p*-value = 0.00013) or insertions of (R = 0.53, *p*-value = 0.017) the same amino acids in the two proteins (Fig. 5f, g) as well as when comparing the average effect of replacing the same amino acids (R = 0.67, *p*-value = 0.012) (Fig. 5e). We note that these correlations are driven by values in the negative range for both peptides, while they are not significant for those mutations increasing nucleation (Supplementary Fig. 12), suggesting that while disruption of amyloid formation proceeds similarly in both peptides, the mechanisms that accelerate nucleation differ between them.

We also evaluated whether these similarities were conserved at a positional level. IAPP and Aβ42 sequences were aligned using T-COFFEE[62] (Fig. 5a), and superimposition of the two protein structures (Fig. 5b) illustrates that they overlap structurally in the fibril core. Across both datasets, 615 substitutions and 649 insertions are found at shared aligned positions in IAPP and Aβ42. Although nucleation scores did not correlate (R = 0.11 for substitutions, R = 0.059 for insertions, Supplementary Fig. 13a, b), categorizing mutations as NS+ or NS− revealed some consistent effects. Among substitutions, 269 (~44%) impacted nucleation of both proteins in the same direction, and 248 (~38%) of insertions also influenced nucleation similarly in both proteins. Most of these mutations decrease nucleation in both proteins (188 substitutions and 189 insertions; NS− in IAPP and NS− in Aβ42, FDR = 0.1) and are more frequently found towards the C-terminus, where the cores of mature fibrils for both peptides are located (Fig. 5c, d). In contrast, only a small number of mutations at aligned positions increase nucleation in both proteins (31 substitutions and 19 insertions; NS+ in IAPP and NS+ in Aβ42, FDR = 0.1) (Fig. 5c, d).

In addition, we find that aggregation and secondary structure predictors (Zyggregator[63], TANGO[64], Camsol[65], s4pred[66]) and variant effect predictors (AlphaMissense[67] and PopEVE[68]) perform poorly when predicting the effects of mutations on nucleation scores in both IAPP and Aβ42[41] (Supplementary Fig. 14).

## Comparing nucleation scores to allele frequency in the population and in diabetes patients

50 IAPP substitutions have been identified in the human population, but their clinical significance remains unknown[69]. In this work, we quantified the nucleation of 49 of them: 29 NS−, 9 NS+, and 11 WT-like

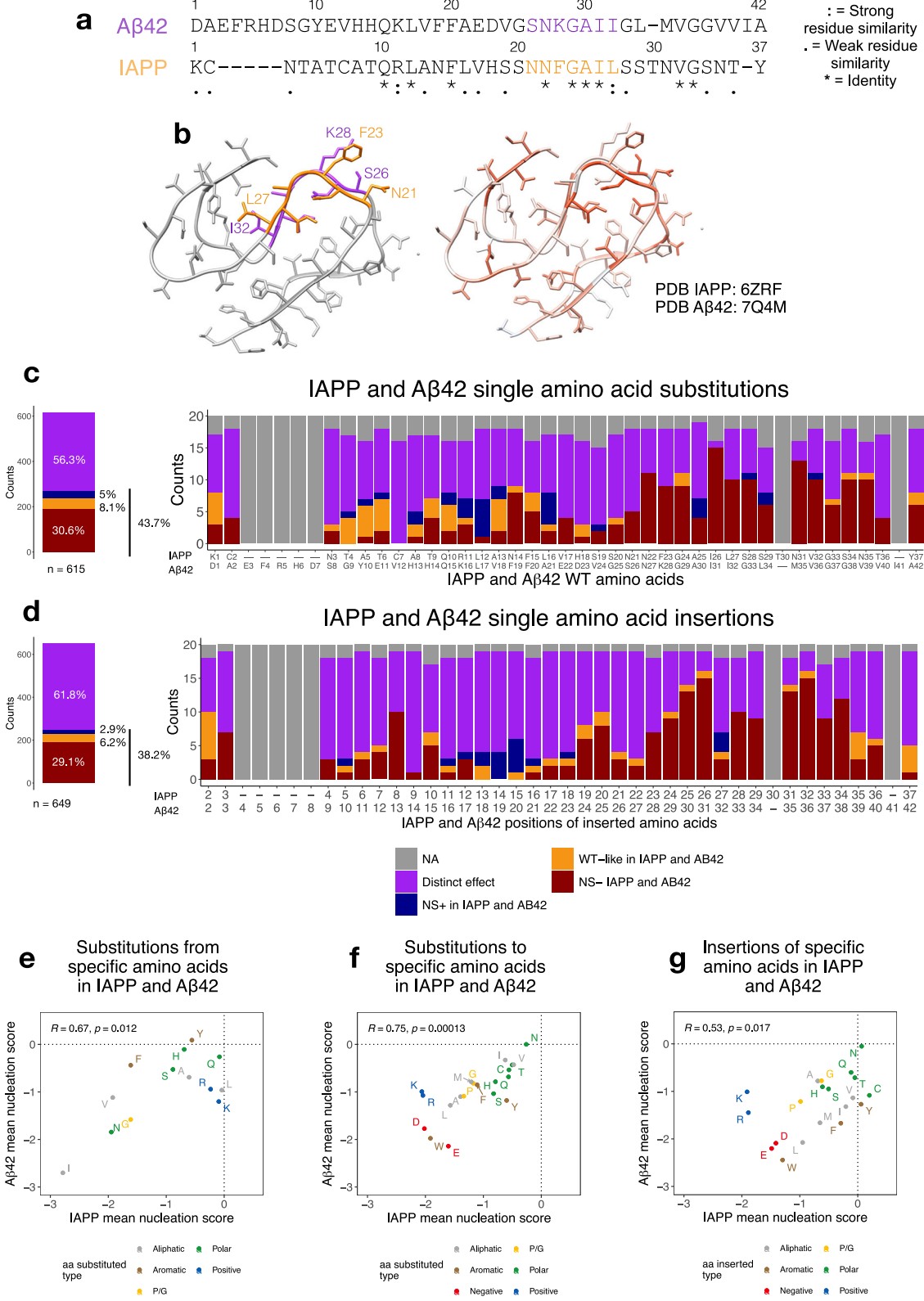

(Supplementary Fig. 15, FDR = 0.1). The variant with the highest allele frequency, p.Ser20Gly, has been extensively characterized in terms of its association with T2D and in vitro aggregation with overall inconclusive evidence[13,70–73] (Supplementary Data 1). We find that, when it comes to amyloid nucleation, this sequence moderately increases nucleation compared to WT (Fig. 2b, Supplementary Fig. 2a).

We also probed whether IAPP variants for which we measure increased amyloid formation could contribute to T2D diabetes by employing the UK Biobank database, which collects whole-genome sequencing and health parameters for 500,000 individuals. We retrieved 28 IAPP variants from the database[74]: 4 NS+, 17 NS−, and 7 WT-like. We find only one NS− variant in individuals with a diabetes

**Fig. 5 | Comparison of mutational effects of IAPP and Aβ42. a** Sequence alignment of IAPP and Aβ42 sequence by T-COFFEE[85]. WT positions of each protein are indicated above its sequence, and gaps are indicated by "-". Conservation scores are indicated below the alignment: "*", ":", "." indicate identical amino acids, conservative changes, and semi-conservative changes, respectively. **b** Structural superposition of IAPP (PDB: 6ZRF[23]) and Aβ42 (PDB: 7Q4M[24]). Structures are colored by the median mutational effect of amino acid substitutions per position (left). The proposed inner core of the fibrils is highlighted for each protein (right).

**c** FDR categories (FDR = 0.1) of nucleation scores of single amino acid substitutions and **d** insertions in IAPP and Aβ42 merged according to T-COFFEE[85] sequence alignment. Missing variants in one or both datasets are colored in gray. The x-axis indicates the WT position and sequence of IAPP and Aβ42. **e** Correlation of the mean nucleation scores of substitutions of specific amino acids, **f** substitutions to specific amino acids, and **g** insertions of specific amino acids in IAPP and Aβ42. Correlation coefficients ($R$) and $p$-values in (**e**), (**f**, and (**g**) were calculated using a two-sided Pearson correlation test.

diagnosis and none in those with glycated hemoglobin (HbA1c) levels above 48 mmol/mol, a standard metric to diagnose and monitor diabetes[75] (Fig. 6a, b). We also employed burden testing to perform rare variant analysis (Sequence Kernel Association Test (SKAT) and odds ratio). Individuals carrying NS− variants have lower odds of receiving a diabetes diagnosis (OR = 0.25, 95% CI 0.03−0.91, $p$ = 0.033) and of presenting elevated HbA1c levels (OR = 0.12, 95% CI 0−0.86, $p$ = 0.029), suggesting these variants may be associated with lower diabetes risk. On the other hand, while we find odds ratios above 1 for NS+ variants (diabetes diagnosis OR 2.27, 95% CI 0.46−6.98, $p$-value: 0.271; elevated HbA1c levels OR 3.9, 95% CI 0.79−11.99, $p$-value: 0.087), confidence intervals are too wide to claim association with increased T2D risk (Fig. 6c, d, Supplementary Data 4). This remains to be determined with an increasing sample size.

## Discussion

Here, we present a comprehensive mutational landscape of IAPP amyloid nucleation, covering 699 single amino acid substitutions, 723 single amino acid insertions, 50 2-amino acid insertions consistent with polymerase slippage, and 443 single and multi-amino acid deletions. This atlas reveals 516 GOF mutations that are able to speed up the aggregation of IAPP. These fast nucleators result from mutations of all types, including insertions, deletions, and substitutions out of the structured core of IAPP amyloids, which would have been hard to predict using currently available aggregation predictors or variant effect predictors.

It would have been difficult to predict these GOF variants even on the basis of the complete mutational landscape of Aβ42, another peptide that aggregates extracellularly and that can, under specific circumstances, form fibrils that structurally align with those formed by IAPP. This suggests that simply one dataset is not enough to recapitulate GOF mutational effects in another amyloid dataset and highlights the need to generate variant effect maps for all human amyloids. Similarities between the two datasets are limited to those mutations that disrupt amyloid nucleation, where changes in nucleation caused by the same types of mutations correlate strongly between datasets. Additionally, in both Aβ42 and IAPP, two interfaces build the structured core of mature fibrils, where strands face each other in a steric zipper. Our results reveal that, for both peptides, mutations are more disruptive in one of the two interfaces.

GOF mutations are found more frequently outside of these core regions for both peptides, including those parts of the sequence that are not even resolved in the final structures of mature amyloid fibrils (1−12 for Aβ42 and 1−11 for IAPP). More specifically, in IAPP, we find that 27.4% of single amino acid substitutions and insertions up to residue 16 are GOF, and identify a region (aa 11−21) where 131 out of 228 (57.4%) insertions speed up nucleation, hinting at the presence of a secondary structure element that in the WT could protect against aggregation by stabilizing the IAPP monomeric ensemble.

While preventively quantifying amyloid nucleation for 1916 IAPP variants in one unique set-up, our study comes with a series of limitations. Although we verified expression levels for a representative set of variants spanning distinct nucleation scores and found no obvious relationship between expression and nucleation, we cannot fully exclude that altered expression may contribute to GOF for other

variants. In addition, given that nucleation takes place in the yeast cytoplasm, it is likely that we are only tracking mutational impact on the reduced version of the peptide. The N-terminal cysteine residues are not part of the published structures of IAPP fibrils, with the exception of the most recent structure of fibrils obtained from T2D patients, where both cysteines are resolved, oxidized, yet do not constitute part of the amyloid core[76]. Several in vitro studies have shown that IAPP forms amyloids in vitro, both in its reduced and oxidized state[77]. However, our results revealed that mutating cysteines and mutation to cysteines have drastic effects, decreasing and increasing nucleation, respectively; we cannot exclude that in this specific experimental set-up cysteines favor nucleation by other mechanisms[78] which include multimerization via metal chelation. However, there are different lines of evidence suggesting our method accurately captures IAPP amyloid nucleation: (1) nucleation scores discriminate significant differences amongst IAPP sequences from different animals, (2) nucleation scores match with most previous in vitro experiments performed in different conditions on unfused peptides and (3) the pattern of mutational impact identifies a core region which corresponds to the minimal IAPP fragment essential for amyloid formation in vitro.

The lack of IAPP variants clearly classified and reported on Clinvar and the limited number of IAPP variants currently available in the UK Biobank make it challenging to validate this dataset with human genetics in a conclusive manner. While we find that individuals carrying variants that decrease nucleation have slightly lower odds of being diagnosed with T2D, the genetic validation of the dataset, unlocking its use to contribute to clinical variant classification, will have to wait for additional data from the sequencing of new T2D patients. However, the preventive quantification of amyloid nucleation presented here could contribute, together with other parameters, to the classification of these variants whenever they are encountered in the clinic or in healthy individuals. There is also a critical need for the design of soluble IAPP analogs. The dataset presented here can help guide the design of IAPP variants that do not aggregate into amyloids, but that are still active and more bioavailable than the currently employed pramlintide.

Finally, this work highlights the importance of developing selection assays that, combined with deep mutagenesis, allow the characterization of gain-of-function variants, which are particularly hard to predict and to characterize, yet are important drivers of human disease.

## Methods
### Ethics

This research has been conducted using the UK Biobank under Application Number 441451. UK Biobank is a large biomedical database containing genetic, imaging, behavioral, and health information from 502,505 participants (229,122 males, 45.6%; 273,383 females, 54.4%) aged 38−73 years living in the UK. Information on sex was based on the data field ID: p22001 (Genetic sex). UK Biobank has approval from the North West - Haydock Research Ethics Committee (REC reference21: /NW/0157). In accordance with this approval, separate ethical approval was not required for our study. All UK Biobank study participants gave written informed consent. The research conformed to the principles of the Declaration of Helsinki.

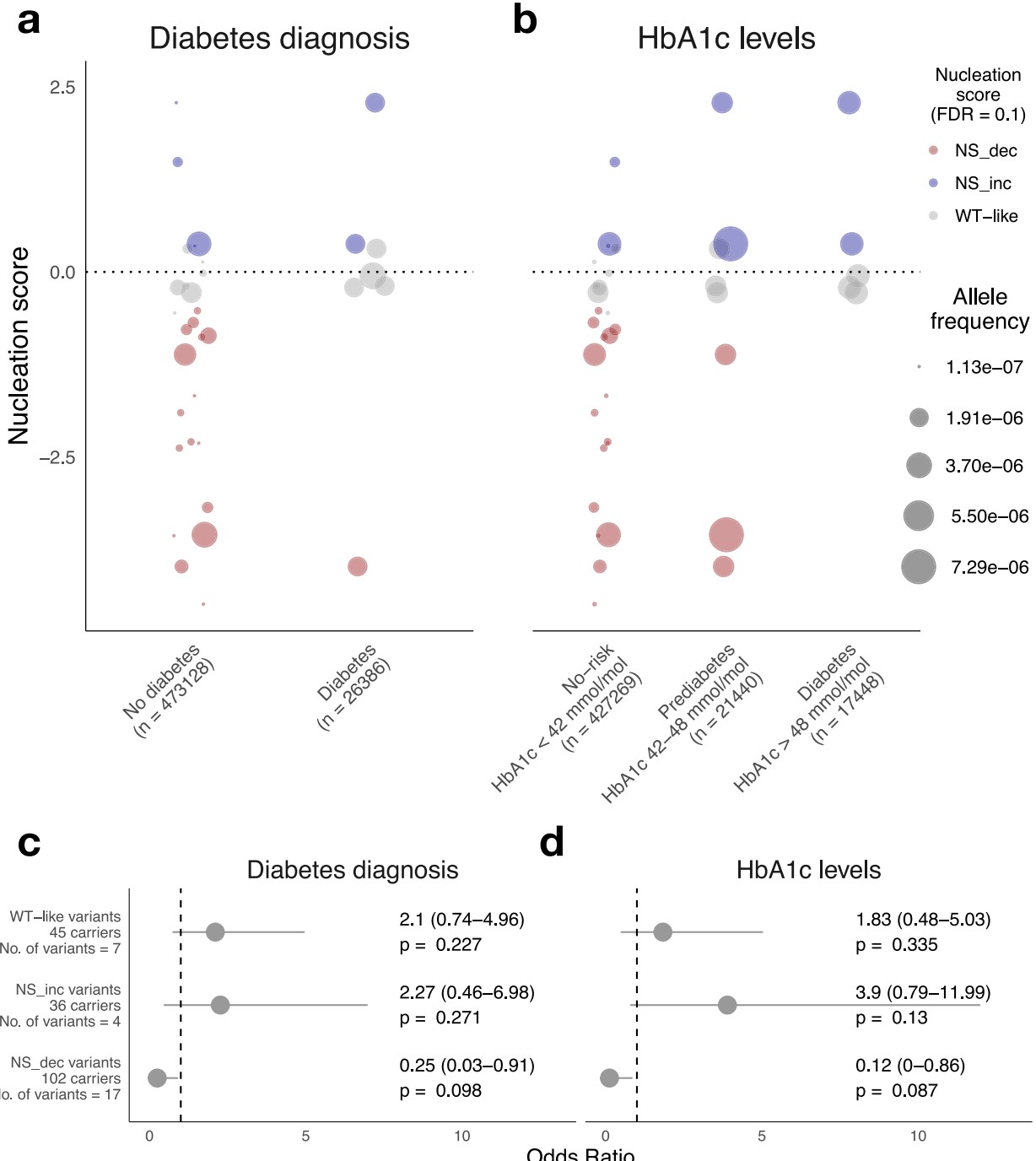

**Fig. 6 | Effect of IAPP variants on T2D diabetes risk. a** Comparison of nucleation scores (y-axis) and allele frequencies of IAPP variants in participants grouped by diabetes diagnosis and **b** by HbA1c levels. Variants are colored according to their effect on IAPP nucleation (NS+, WT-like, NS−, FDR = 0.1). Point size represents the allele frequency of each IAPP variant in each group. **c** Forest plots showing odds ratio (points), 95% confidence interval, and Benjamini–Hochberg adjusted two-sided *p*-values from Firth penalized logistic regression models for diabetes diagnosis across IAPP variant groups, adjusted for age, BMI, and sex, and **d** corresponding analyses for HbA1c levels. The number of variants and carriers in each group is indicated.

## Library design

Two independent IAPP libraries were assayed, each with three biological replicates. Library 1 (Twist Bioscience) was designed to contain 2000 unique variants: all possible single amino acid substitutions and stop codons (*n* = 738), single amino acid insertions (*n* = 685), single amino acid deletions (*n* = 32), internal deletions of 2–16 residues (*n* = 374), N- and C-terminal truncations (*n* = 27 each), 73 polymerase-slippage variants, 43 synonymous variants, and the wild-type

sequence. Because coverage of single substitutions and insertions in Library 1 was incomplete, Library 2 (Integrated DNA Technologies) was produced using an NNK-based design to increase representation of these mutations (*n* = 2400). All variants are numbered relative to the mature human IAPP peptide (residues 1–37), which corresponds to positions 34–70 of the canonical human preproIAPP sequence (NP_000406.1). Amino acid substitutions, insertions, and deletions are annotated in HGVS-compliant protein-level notation relative to this

mature sequence (e.g., p.Ser20Gly). Full HGVS notations for variants mentioned in the text, including a mapping to preproIAPP positions, are provided in Supplementary Data 2.

## Plasmid library construction

Both IAPP libraries were processed following the same protocol. Synthetic oligonucleotide pools were obtained from Twist Biosciences (Library 1) or Integrated DNA Technologies (Library 2), with IAPP variant regions ranging from 30 to 117 nt, flanked by 25 nt upstream and 21 nt downstream constant regions. 10 ng of the library were amplified by PCR (Q5 high-fidelity DNA polymerase, NEB) for 10 cycles with primers annealing to the constant regions (primers MB_01-02, Supplementary Data 3), following the manufacturer's protocol. The products were treated with 2 μL of ExoSAP (Affymetrix) and then purified by column purification (MinElute PCR Purification Kit, Qiagen). The pCUP1-Sup35N plasmid[43] was linearized by PCR (Q5 high-fidelity DNA polymerase, NEB) (primers MB_03-04, Supplementary Data 3). The product was treated with DpnI (Thermo Fisher Scientific) and purified from a 1% agarose gel (QIAquick Gel Extraction Kit, Qiagen).

Each library was ligated to 100 ng of the linear vector with a ratio of 5:1 (library:vector) by a Gibson approach with 3 h of incubation. The reaction products were dialyzed for 30 min on a membrane filter (MF-Millipore 0.025 μm membrane, Merck) and transformed into 10-beta Electrocompetent *E. coli* (NEB), by electroporation with 2.0 kV, 200 Ω, 25 μF (BioRad GenePulser machine). Cells were recovered in SOC medium for 30 min and grown overnight in 50 ml of LB ampicillin medium. A number of cells were plated in LB ampicillin plates to calculate the transformation efficiency and confirm that each variant in the library would be represented at least 10 times. A total of approximately 116,000 and 1.5 M transformants were obtained for Library 1 and Library 2, respectively. A total of 2 ml of overnight culture was harvested to obtain a miniprep of each of the libraries (NZYMiniprep kit, NZYTech).

## Yeast transformation

*Saccharomyces cerevisiae* [psi-][pin-] (MATa ade1-14 his3 leu2-3,112 lys2 trp1 ura3-52) strain was used in all experiments in this study. Yeast cells were transformed with each of the two IAPP plasmid libraries, in three independent biological replicates per library. An individual colony was grown overnight in 3 ml of YPDA medium at 30 °C and 400 × g. Cells were diluted to OD600 = 0.3 in 60 ml of YPDA medium and grown at 30 °C and 400 × g for 4 h. Cells were harvested and split into 4 transformation tubes of 15 ml each. Each tube was treated as follows: cells were harvested at 400 × g for 5 min and washed in 1 ml of YTB (100 mM LiOAc, 10 mM Tris, pH 8, 1 mM EDTA). They were harvested again and resuspended in 72 μl YTB. 150 ng of the corresponding IAPP plasmid library, 8 μl of previously boiled ssDNA 10 mg/uL (UltraPure, Thermo Scientific), 60 μl of DMSO, and 500 μl of YTB + PEG (100 mM LiOAc, 10 mM Tris pH 8, 1 mM EDTA pH 8, 40% PEG3350) were added to the cells. Heat shock was performed for 14 min at 42 °C in a thermo block. Finally, cells were harvested and resuspended, and the 4 transformation tubes were pooled and added to a conical flask with 50 ml plasmid-selection medium (-URA, 2% glucose). Cells were grown for 50 h at 30 °C and 400 x g. Transformation efficiency was determined for each biological replicate by plating an aliquot of the transformed cells on plasmid-selection agar. For Library 1, an estimated 345,000, 102,000, and 264,000 transformants were obtained across the three biological replicates, corresponding to a minimum per-variant coverage of 51-fold. For Library 2, the three replicates yielded 181,500, 168,500, and 143,000 transformants, providing at least 59-fold coverage per variant. After 50 h of growth, each culture was diluted to OD600 = 0.05 in 50 ml plasmid-selection medium and grown to OD600 = 0.8–1. Cells were harvested and stored at −80 °C in 25% glycerol.

## Selection assays

Selection assays were performed independently for each library. For Library 1, each biological replicate was assayed in two technical replicates, whereas Library 2 was assayed with one technical replicate per biological replicate.

For each replicate, cells were thawed from −80 °C in 50 ml plasmid-selection medium at OD = 0.05 and grown until exponential for 15 h. At this stage, cells were harvested and resuspended at OD = 0.1 in 50 ml protein induction medium (-URA, 2% glucose, 100 μM Cu$_2$SO$_4$). After 24 h, input pellets were collected (10 ml per technical replicate) and stored at −20 °C for later DNA extraction. For the selection step, approximately 135 million cells per replicate were plated on −ADE − URA medium for Library 1 and 72 million for Library 2 (145 cm² plates; Nunc, Thermo Scientific). Plates were incubated at 30 °C for 6 days inside an incubator. Finally, colonies were scraped off the plates with PBS 1× and harvested by centrifugation to collect the output pellets, which were stored at −20 °C for later DNA extraction.

## DNA extraction and sequencing library preparation

One input and one output pellet were collected for each biological replicate (and technical replicate in the case of Library 1) of each of the libraries. Pellets were resuspended in 0.5 ml extraction buffer (2% Triton X-100, 1% SDS, 100 mM NaCl, 10 mM Tris-HCl, pH 8, 1 mM EDTA, pH 8). They were then frozen for 10 min in an ethanol-dry ice bath and heated for 10 min at 62 °C. This cycle was repeated twice. 0.5 ml of phenol:chloroform:isoamyl (25:24:1 mixture, Thermo Scientific) was added together with glass beads (Sigma). Samples were vortexed for 10 min and centrifuged for 30 min at 20,000 × g. The aqueous phase was then transferred to a new tube, and mixed again with 0.5 ml of phenol:chloroform:isoamyl, vortexed, and centrifuged for 45 min at 20,000 × g. Next, the aqueous phase was transferred to another tube with 1:10 V 3 M NaOAc and 2.2 V cold ethanol 96% for DNA precipitation. After 30 min at −20 °C, samples were centrifuged, and pellets were dried overnight. The following day, pellets were resuspended in 0.3 ml TE 1× buffer and treated with 10 μl RNAse A (Thermo Scientific) for 30 min at 37 °C. DNA was finally purified using 10 μl of silica beads (QIAEX II Gel Extraction Kit, Qiagen) and eluted in 30 μl elution buffer. Plasmid concentrations were measured by quantitative PCR with SYBR green (Merck) and primers annealing to the origin of the replication site of the pCUP1-Sup35N-IAPP plasmid at 58 °C for 40 cycles (primers MB_05-MB_06, Supplementary Data 3).

Libraries were prepared for high-throughput sequencing using a two-step PCR protocol (Q5 high-fidelity DNA polymerase, NEB). In PCR1, 30 million molecules were amplified for 15 cycles with frameshifted primers with homology to Illumina sequencing primers (primers MB_07-MB_20, Supplementary Data 3). The products were treated with ExoSAP (Affymetrix) and purified by column purification (MinElute PCR Purification Kit, Qiagen). They were then amplified for 10 cycles in PCR2 with Illumina-indexed primers (primers MB_21-MB_28, Supplementary Data 3). The six samples (3 inputs and 3 outputs) of each library were pooled together equimolarly, and the final product was purified from a 2% agarose gel with 20 μl silica beads (QIAEX II Gel Extraction Kit, Qiagen).

The libraries were sequenced using 125 bp paired-end reads on an Illumina NextSeq2000 at the CRG Genomics Core Facility. Sequencing yielded more than 65 million paired-end reads for each of the two libraries. This corresponds to 1.3–6.3 million reads per sample for Library 1 and 7.8–11.25 million per sample for Library 2 (i.e., input or output for a specific technical and biological replicate), representing >665× read coverage for each designed variant in Library 1 and >3250× in Library 2.

## Individual variant testing

Selected IAPP single substitutions for individual testing were obtained by PCR linearization of the pCUP1-Sup35N-IAPP plasmid

(Q5 high-fidelity DNA polymerase, NEB) with mutagenic primers (primers MB_28-MB_57, Supplementary Data 3). IAPP animal variants were obtained by ultramer amplification (primers MB_58-60 and MB_01-02, Supplementary Data 3) and Gibson assembly with the pCUP1-Sup35N-IAPP linearized plasmid (primers MB_01-02, Supplementary Data 3). PCR products were treated with DpnI overnight and transformed into DH5 alpha-competent *E. coli*. Plasmids were purified by miniprep (NZYMiniprep kit, NZYTech) and transformed into yeast cells using one transformation tube of the transformation protocol described above. All constructions were verified by Sanger sequencing. Plasmids containing selected IAPP substitutions were then linearized (primers MB_04&MB_61, Supplementary Data 3) and used as templates for assembling the hemagglutinin (HA) tag by Gibson assembly using primers MB_62-65 (Supplementary Data 3). The resulting constructs were transformed into DH5α-competent *E. coli*, purified by miniprep, verified by Sanger sequencing, and subsequently transformed into yeast cells as described above.

To determine the % growth in an adenine-lacking medium, yeast cells expressing individual variants were grown overnight in plasmid-selection medium until exponential (-URA, 2% glucose). Protein expression then was induced by diluting the cells to an OD of 0.05 in protein induction medium (-URA, 2% glucose, 100 μM Cu$_2$SO$_4$) and grown for 24 h. Cells were plated on -URA (control) and -ADE-URA (selection) plates in three independent replicates, and grown for 6 days at 30 °C. Growth in an adenine-lacking medium was calculated as the percentage of colonies in -ADE-URA relative to colonies in -URA.

For individual growth rate measurements, yeast cells expressing individual variants were grown overnight in plasmid-selection medium (-URA 2% glucose) and diluted to an OD of 0.1 until exponential. They were then diluted again to OD 0.08 in non-inducing (-URA 2% glucose) and inducing (-URA 2% glucose 100 μM Cu$_2$SO$_4$) protein expression media. Cell growth was measured at 30 °C for >48 h at 10 min intervals in a microplate reader (SPARK, Tecan) in three biological replicates. Growth rates were calculated as the maximum slope of the linear fit of log(OD) over time at the exponential phase of the growth curve.

### Growth assays with fused and unfused Sup35N-IAPP constructs

To generate unfused IAPP expression constructs, the p415GAL1-empty plasmid (gift from the Lehner lab) was linearized by PCR (primers MB_65-66, Supplementary Data 3), and the IAPP coding sequence was amplified from pCUP-Sup35N-IAPP using Q5 High-Fidelity DNA Polymerase (NEB) (primers MB_67-68, Supplementary Data 3). PCR products were treated with DpnI (Thermo Fisher Scientific), purified (QIAquick and MinElute PCR Purification Kits, Qiagen), and assembled by Gibson assembly. Plasmids were transformed into DH5α-competent *E. coli*, isolated by miniprep, and verified by Sanger sequencing.

To generate the pCUP-empty control, pCUP-Sup35N (gift from the Chernoff lab) was linearized by PCR to remove the Sup35 N-terminal domain (primers MB_69-70, Supplementary Data 3), followed by DpnI treatment, column purification (QIAquick PCR Purification Kit, Qiagen), transformation into DH5α-competent *E. coli*, and sequence verification.

Appropriate p415GAL1 and pCUP plasmids encoding fused or unfused Sup35N-IAPP constructs were co-transformed into yeast cells as described above. Transformants were grown overnight in −URA −LEU medium containing 2% glucose, diluted to OD$_{600}$ = 0.05 in induction medium (−URA −LEU, 2% galactose, 100 μM Cu$_2$SO$_4$), and incubated for 24 h. Cells were adjusted to OD$_{600}$ = 1, serially diluted 10-fold, and spotted onto −URA −LEU (control) and −ADE −URA −LEU (selection) plates. Plates were incubated at 30 °C for 6 days before imaging.

### Thioflavin-T binding assay

Synthetic purified peptides were purchased from Bachem as TFA salts. The peptides were resuspended in TFA, sonicated, frozen, and lyophilized. After HFIP treatment, peptides were stored at −80 °C. For the Thioflavin-T (ThT) assay, peptides were resuspended in 50 mM NaP buffer (pH = 7.4) with 20 μM ThT. Fluorescence was measured at 480 nm every 5 min at 29 °C. Peptide concentrations were determined by acid hydrolysis and amino acid analysis by the Separative Techniques Unit of the Scientific and Technological Centers of the University of Barcelona (CCiTUB).

### Protein extraction and Western blotting

Yeast strains expressing individual HA-tagged IAPP variants were grown overnight in plasmid-selective medium (−URA) containing 2% glucose until mid-exponential phase. Protein expression was induced by diluting cultures to an OD600 of 0.05 into induction medium (−URA, 2% glucose) supplemented with 100 μM Cu$_2$SO$_4$ and incubating for 24 h. For harvest, 50 mL of culture at OD600 = 2.5 was collected by centrifugation at 3000 × *g* for 5 min at 4 °C, washed once with 1 mL ice-cold PBS, and pelleted by centrifugation at 500 × *g* for 5 min at 4 °C. Cell pellets were resuspended in 400 μL ice-cold lysis buffer (25 mM Tris-HCl, pH 7.4; 100 mM NaCl; 1 mM EDTA; 2.5% (v/v) glycerol; 0.5% (v/v) Triton X-100; 0.25% (w/v) sodium deoxycholate; 0.05% (w/v) SDS; 0.5 mM DTT; 1 mM PMSF; 1× cOmplete™ EDTA-free protease inhibitor (Roche)) and mixed with ~200 μL acid-washed glass beads. Cells were lysed by vortexing six times for 30 s with ≥1 min on ice between pulses. Lysates were clarified by centrifugation at 10,000 × *g* for 10 min at 4 °C, and the supernatant was retained as the total protein extract.

Protein samples were combined with denaturing/loading buffer (8 M urea; 4% SDS; 20% glycerol; 120 mM Tris-HCl, pH 7.4; 10% β-mercaptoethanol; bromophenol blue), heated at 50 °C for 10 min, and separated on Bis-Tris 4–12% polyacrylamide gels (NZYTech). Proteins were transferred to 0.2 μm nitrocellulose membranes (iBlot™ 2 Transfer Stacks, Thermo Fisher Scientific). Membranes were blocked for 30 min in 5% (w/v) nonfat dry milk in PBS and incubated overnight at 4 °C with primary antibodies: anti-HA (ChIP grade, AB9110, Abcam; 1:1000) and anti-GAPDH (loading control, AB9484, Abcam; 1:5000) diluted in 1% (w/v) nonfat dry milk in PBS. Membranes were washed six times with PBS containing 0.1% (v/v) Tween-20. Membranes were then incubated for 50 min at room temperature with HRP-conjugated secondary antibodies appropriate to the primary species (anti-rabbit IgG−HRP, GENA934, Cytiva; anti-mouse IgG−HRP, sc-516102, Santa Cruz Biotechnology), each at 1:5000 in PBS, followed by six additional washes with PBS-0.1% Tween-20. Immunodetection was performed using Amersham ECL™ Prime Western Blotting Detection Reagent according to the manufacturer's instructions and imaged on a chemiluminescence system.

### Data processing

FastQ files from paired-end sequencing of the IAPP library were processed using DiMSum[79] (available at: https://github.com/lehner-lab/DiMSum), an R pipeline for analyzing deep mutational scanning data. 5′ and 3′ constant regions were trimmed, allowing a maximum of 20% of mismatches relative to the reference sequence. Sequences with a Phred base quality score below 30 were discarded. Non-designed variants and variants with fewer than 100 input reads in Library 1 and fewer than 50 in Library 2 across the 3 biological replicates were also excluded from further analysis. Filtering thresholds were chosen based on estimates provided by DiMSum.

### Nucleation scores and error estimates

Nucleation scores (NS) and their error estimates were calculated using the DiMSum package (https://github.com/lehner-lab/DiMSum)[79] for each variant in each biological replicate. For each variant ($i$) in replicate ($r$), an enrichment score (ES) was first computed as the natural logarithm of the ratio between its normalized sequencing read counts in the output (collected after selection) and input (collected before selection) samples: ES$_{i,r}$ = log($F_{i,r}$ output) − log($F_{i,r}$ input), where $F$ represents the sequencing read frequency of a given variant in each

library. Likewise, the enrichment score for the IAPP WT sequence was calculated as: $ES_{WT,r} = \log(F_{WT,r}$ output$) - \log(F_{WT,r}$ input$)$. The nucleation score for each variant was then defined as the difference between its enrichment score and that of the WT in the same replicate: $NS_{i,r} = ES_{i,r} - ES_{WT,r}$.

For each library, nucleation scores of individual variants were first merged across biological replicates using an error-weighted mean and centered to the mean NS of wild-type IAPP and synonymous variants within the same library. To integrate the two libraries, we applied Deming regression to variants shared between Library 1 and Library 2. The resulting slope and intercept were used to adjust the NS values (fitness_merged) and associated standard errors (sigma_scaled) of variants not present in Library 2, aligning them to the Library 2 scale. Variants present in Library 2 retained their original, untransformed values. Variants unique to Library 1 were then added to the merged dataset after this rescaling. This procedure produced a single harmonized NS dataset used for all downstream analyses and provided in Supplementary Data 2. Raw variant counts and per-library nucleation score estimates prior to library integration are provided in Supplementary Data 5 (Library 1) and Supplementary Data 6 (Library 2).

## Data analysis

**Variants in the library.** NS was obtained for 1916 unique IAPP variants, which were split into mutation classes: 699 single amino acid substitutions, 723 single amino acid insertions, 25 single amino acid deletions, 358 internal multi-amino acid deletions, 50 variants resulting from polymerase slippage, 60 truncations (from either N-terminal or C-terminal), and WT IAPP.

For single amino acid insertions, the position of the inserted residue is assigned relative to the surrounding peptide sequence. For example, an insertion between positions 1 and 2 is labeled as position 2. Insertions before the first residue of the mature peptide are labeled as position 1 in the figures (e.g., an insertion of serine before the first residue is shown as p.0_Lys1insSer), while insertions after the last residue (position 37) are labeled as position 38 in the figures (e.g., an insertion of leucine after residue 37 is shown as p.Tyr37_insLeu).

Different mutations can result in the same coding sequence (e.g., Δ19–27, Δ20–28, and Δ21–29, which are all the same protein sequence: KCNTATCATQRLANFLVHSSTNVGSNTY). This is the case for single amino acid insertions, truncations, and single and multi-amino acid deletions. They are only considered as one coding variant but are considered multiple times for visualization or if the analysis is position-specific, in figures: Fig. 4 and Supplementary Figs. 6, 8, 9, 10, and 11.

To analyze the impact of single amino acid insertions on IAPP nucleation, we calculated the frequency of mutations per position that increased or decreased nucleation scores. A contiguous stretch of positions where NS+ insertions were the most frequent defined the central region (aa 11–21), while the remaining positions were assigned to N-terminal (aa 1–10) and C-terminal (aa 22–38) regions, where NS+ insertions were less common.

**Aggregation, amyloid nucleation, and variant effect predictors.** For the aggregation, solubility, and secondary structure predictors (TANGO[64], Zyggregator[63], Camsol[65], and s4pred[66]), individual residue-level scores were summed to obtain a score for each IAPP variant and normalized by the sequence length. Alpha- and beta-propensity from TANGO were used for correlation. For the "No. of helix predicted residues" value, residues predicted as helical in s4pred[66] were counted for each IAPP variant and normalized by the sequence length.

For PopEVE[68], we downloaded the pre-computed popEVE scores for IAPP from the PopEVE server and used the "pop.adjusted_EVE" and the "ESM1v" values for correlation. For AlphaMissense[67], the pre-computed pathogenicity scores were obtained from the AlphaMissense server.

Principal components from a previously published PCA[80] that reduced redundancy of amino acid physicochemical properties were also used for correlation.

**Correlations of relative accessible surface area and nucleation scores.** Relative accessible surface area (ASA) of each residue in IAPP WT PDB structures (PDB IDs: 7M61[21], 7M62[21], 7M64[21], 7M65[21], 6Y1A[14], 6ZRF[23], 8R4I[19], and 6VW2[20]) was obtained using the DSSP package from BioPython[81] and correlated to the mean NS of each IAPP position. Residues are classified as buried if their relative ASA is <0.25[82].

**Comparison of gnomAD and UK BioBank allele frequencies and nucleation scores.** We extracted allele frequencies of IAPP variants (ENST00000240652.8, NM_000415.3) in the human population from gnomAD v4.1.0[69].

Self-reported diabetes status, HbA1c levels, age, sex, and body mass index were extracted for 501,252 UK Biobank participants, including 192 who carried a non-synonymous IAPP variant. Participants from the UK Biobank were classified into three different groups based on their HbA1c levels, according to the WHO guidelines for diabetes diagnosis: (i) no-risk: participants with HbA1c levels lower than 42 mmol/mol ($n = 427269$), (ii) pre-diabetes: participants with HbA1c levels between 42 and 48 mmol/mol ($n = 21440$) and (iii) diabetes: participants with HbA1c levels greater than 48 mmol/mol ($n = 17448$). No-risk and pre-diabetes participants were considered as controls, and diabetes participants were considered as cases. Participants with HbA1c levels outside the analytical range of the instrument (15–184 mmol/mol) were excluded from the analysis. In parallel, participants were categorized on the basis of their response to the question: "Has a doctor ever told you that you have diabetes?" during their first visit as follows: (a) diabetes: participants who answered "YES" ($n = 26386$) and (b) no diabetes: participants who answered "NO" ($n = 473128$). To assess the association of IAPP variants and diabetes diagnosis, participants in (a) and (b) were considered cases and controls, respectively. Participants who didn't select either option were excluded from the analysis. Allele frequencies for each group were retrieved from the UK Biobank for comparison to nucleation scores.

Association between IAPP variants and diabetes diagnosis or HbA1c levels was assessed using the SKAT R package[83], with age, sex, and body mass index included as covariates. Analyses were performed either on all IAPP variants present in the UK Biobank or on variants grouped according to their effect on nucleation (NS+, WT-like, and NS−). To obtain robust $p$-values, we performed permutation testing ($n = 1000$), in which phenotypes were randomly shuffled across individuals while preserving the genotype structure. This approach accounts for potential deviations from the asymptotic assumptions of SKAT, particularly in groups with a small number of variants, rare variants, or highly skewed effect sizes. Both asymptotic and empirical $p$-values for burden, SKAT, and SKAT-O tests are reported in Supplementary Data 4.

Odds ratios were calculated using multivariable logistic regression with Firth's penalized likelihood method, adjusting for age, sex, and body mass index. Variants were grouped according to their nucleation effect (NS+, WT-like, and NS−) for these analyses.

## Statistics and reproducibility

Based on the transformation efficiency, each variant in the designed libraries is expected to be represented at least 50× at each step in the selection experiments and library preparation. In terms of sequencing, reads that did not pass the QC filters using the DiMSum package were excluded (https://github.com/lehner-lab/DiMSum). The experiments were not randomized. The investigators were not blinded to allocation during experiments and outcome assessment.

**Reporting summary**

Further information on research design is available in the Nature Portfolio Reporting Summary linked to this article.

## Data availability

Raw sequencing data are deposited in the European Nucleotide Archive (ENA) under project accession number: PRJEB104038. The processed data (nucleation score estimates and associated error terms) are provided in Supplementary Data 2 and have also been deposited in MaveDB[84], a dedicated repository for standardized deep mutational scanning data. The processed dataset can be found by searching *IAPP* from the MaveDB landing page [https://www.mavedb.org/]. The Universal Resource Name (URN) for the dataset is urn:mavedb:00001253-a [https://www.mavedb.org/experiments/urn:mavedb:00001253-a]. The amyloid beta 42 nucleation dataset was obtained from Zenodo [https://doi.org/10.5281/zenodo.7255570]. The coordinates for the PDB structures used in the study were obtained with accession numbers 7M61[21], 7M62[21], 7M64[21], 7M65[21], 6Y1A[14], 6ZRF[23], 8R4I[19], 6VW2[20], 8AWT[22], 8AZ0[22], 8AZ1[22], 8AZ2[22], 8AZ3[22], 8AZ4[22], 8AZ5[22], 8AZ6[22], 8AZ7[22], 6ZRR[23], 6ZRQ[23], 7Q4M[24]. Variant frequency data were obtained from gnomAD v4.1.0[69] [https://gnomad.broadinstitute.org/gene/ENSG00000121351?dataset=gnomad_r4] and UK Biobank [https://www.ukbiobank.ac.uk/enable-your-research/register]. Source data are provided with this paper.

## Code availability

All analysis scripts are available in the project's GitHub repository [https://github.com/BEBlab/MAVE-IAPP] and in Zenodo [https://doi.org/10.5281/zenodo.18509746].

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

## Acknowledgements

M.B. was supported by the fellowship "Ayudas para contratos predoctorales para la formación de doctores 2019" (PRE2019-088300) from the Spanish Ministry of Science, Innovation and Universities. The project that gave rise to these results has received funding from "la Caixa" Research Foundation under the grant agreement LCF/PR/HR21/52410004 (project 'DeepAmyloids'). Work in the lab of B.B. is also supported by the Spanish Ministry of Science, Innovation and Universities (PID2021-127761OB-I00 and RYC2020-028861-I, funded by MCIN/AEI/10.13039/501100011033, "ERDF A way of making Europe" and "ESF Investing in your future"), and by the European Union (ERC Consolidator, Glam-MAP, 101125484). Views and opinions expressed are, however, those of the author(s) only and do not necessarily reflect those of the European Union or the European Research Council. Neither the European Union nor the granting authority can be held responsible for them. IBEC is a member of the CERCA Program/Generalitat de Catalunya. We thank the Chernoff and the Lehner lab for providing strains and plasmids, and the CRG Genomics core technology for sequencing. We thank Prof. John Perry, Dr. Yajie Zhao, Prof. Ben Lehner, Carla Folgado, and the B.B. lab members for discussing our data.

## Author contributions

B.B. conceptualized and supervised the project. C.B. and M.B. designed the libraries and performed the experiments. M.B. analyzed the data and conducted the UK Biobank analyses. B.B. and M.B. interpreted the results and wrote the manuscript. All authors reviewed and edited the manuscript.

## Competing interests

The authors declare no competing interests.
