## [Transparent Peer Review file · Nature Communications]

Massively parallel quantification of mutational impact on IAPP amyloid formation

Corresponding Author: Dr Benedetta Bolognesi

Version 0:

Reviewer comments:

Reviewer #1

(Remarks to the Author)

The manuscript by Badia, Battle, and Bolognesi focuses on the high-throughput quantification of over 1,600 genetic mutations and their effects on the amyloid aggregation of IAPP. Through their analysis, the authors identify critical residues, particularly the stretch from residue 15 to 32 as essential for the aggregation of the peptide. They also compare these results with previous data on the amyloid-beta peptide. The rationale behind this comparison is that the two peptides share significant sequence similarity. The authors find that the effects of mutations that slow down nucleation correlate between the two amyloids. However, when it comes to mutations that trigger or accelerate nucleation, a dataset from one amyloid cannot be used to predict mutational effects in the other.

This is a highly novel and interesting paper that makes important contributions to both method development and foundational science. The dataset they have obtained will be highly relevant and useful to the protein chemistry community. Overall, the manuscript is well written, and the findings are supported by the data. I believe this work will be of broad interest to the Nature Communications' readership. Thus, I am supportive of its publication, provided that the authors address my following minor concerns.

- 1) The authors employ a Sup35 fusion strategy in which IAPP sequences are fused to Sup35. Since Sup35 nucleation is required for yeast survival in the absence of adenine, variants that aggregate/ form amyloid fibrils in yeast are selectively enriched. The authors should clarify how they control for potential false negatives. There is a scenario in which a variant forms aggregates that are toxic to yeast, causing the cells to die or grow poorly despite successful nucleation. As a result, such variants may appear to have low nucleation potential, leading to false negatives in the selection assay. Given the presence of WT control and the high number of mutants analysed, this is probably not a problem. However, I would encourage the authors to discuss this aspect in the manuscript.
- 2) Similarly, the authors should clarify how they normalize for differences in protein expression levels across variants. For example, if a variant is expressed at lower levels, it might appear to have reduced nucleation potential, even if it can aggregate or form amyloids.
- 3) Could the authors clarify how the analysis in Figure 1e was performed? Specifically, how were these variants selected? Were the aggregation conditions across these papers/published results comparable, or were the results normalized to a control, e.g., the WT IAPP?
- 4) The authors present a compelling analysis by mapping their nucleation scores onto high-resolution structures of amyloid fibrils formed by IAPP or amyloid-beta. By identifying key residues aligning with fibril cores, they provide support that the yeast-formed aggregates are amyloid-like and resemble those described in the literature. To further strengthen this claim, it would be helpful to include additional supporting evidence. For example, could the authors quantify IAPP-positive, SDS-insoluble fractions for a few relevant yeast variants (e.g., mutants within the fibril core and the WT) to assess whether there is any correlation with the nucleation score?
- 5) The authors identify Cys7 to be an important residue for IAPP aggregation. However, it is not involved in a disulphide in amyloid structures from patients. What is the proposed role of this residue?
- 6) Does the nucleation score reflect a thermodynamic property (e.g., stability of the final aggregated state), a kinetic

parameter (e.g., rate of nucleation), or a combination of both?

7) The authors should provide additional information on how the Nucleation Score was calculated in the section Nucleation scores and error estimates. For example, they should clarify what “log(Fi or wt input or output)” represents, to make this clearer to a wider readership. I understand that this refers to the in vivo enrichment and that more details were provided in the original ELife 2021 paper. However, it would be helpful to include more information here to make the manuscript more self-contained.

(Remarks on code availability)

The link provided directs you to the GitHub page of the team. I would recommend that the authors report the link that sends directly to the project's folder. All scripts to make all the figures and tables are present. Instructions on how to install and run the code are also present. Scripts are commented to help the analysis.

Reviewer #2

(Remarks to the Author)

The manuscript presents data derived from deep mutational scanning of the 37-residue polypeptide IAPP, a highly amyloidogenic polypeptide whose amyloid self-assembly is closely associated to T2D pathogenesis. The aim was to identify mutations or other sequence changes that affect the amyloidogenic potential of IAPP by deep mutational scanning. The authors used MAVEs and deep indel mutagenesis methodology (DIM) recently used to study Abeta42 of AD (ref 31). They generated 1663 IAPP variants which comprised variants with single amino acid substitutions, insertions or deletions, and variants with truncations, larger internal deletions, and sequence alterations which could result from polymerase slippage. Amyloidogenic potentials (termed “nucleation score” or NS in comparison to (human) IAPP) were quantified by using the yeast cell assay (ref. 53) which was also used in the Abeta42 studies (ref 53) and which they adapted for studying IAPP.

Overall the results are very impressive and consistent with results by others based on structural models of IAPP fibrils and early aggregates and derived from various structure-amyloidogenicity studies which were done using classical biochemical/biophysical methodology. In summary, the data is consistent with IAPP(15-32) being a core sequence of IAPP fibrils and early aggregates. Also, it is shown that mutations have a more drastic effect in IAPP(21-27) or NNFGAIL which was earlier found to be a key amyloid core segment of IAPP. The approach could find broad applicability in the amyloid field and beyond.

Additional comments

1) The results are derived from a yeast cell assay that is based on the capability of a fusion construct of IAPP with the nucleation domain of Sup35 to nucleate Sup35 amyloid formation which is required for yeast survival under the lack of adenine. The latter one provides a functional readout (NS or nucleation score) for the Sup35 amyloid nucleating ability of IAPP or its diverse variants, which should according to the authors' hypothesis correlate with the amyloidogenic potential of IAPP and its variants as found in the case of analogous Abeta42/Sup35 constructs (ref. 31).

(1a) Regarding the proof-of-principle for the suitability of the cell assay: In Fig. 1b the authors present data related to the suitability of their assay for quantifying amyloidogenic potentials. However, it is unclear whether the differences between the relative growth rates of the different IAPP variants are significant because information about the x-axis (relative growth units and scale (is it a logarithmic scale?)) and the significance of the differences between the data of IAPP and its variants is missing.

(1b) The authors' conclusions are based exclusively on the readout of their yeast cell assay which is an indirect assay. However, I am missing direct biochemical assay validation experiments in the manuscript and there are some important questions on the principle and the applicability of the assay. For instance, which is the nucleating agent? Is this amyloid-like fibrillary assemblies formed by the construct of IAPP(or its amyloidogenic IAPP variant) with Sup35(nucleation domain). Are Sup35 amyloid-like fibrils cross-nucleated by the fibrils of the construct of IAPP(amyloidogenic IAPP)/Sup35(nucleation domain)? What is the effect of kinetics of amyloid-like fibril formation and fibril thermodynamic stability of IAPP(IAPP variants) and of their constructs with Sup35(nucleation domain) on the assay output? What is the effect of potential intrinsic cytotoxicity of IAPP-related aggregates and fibrils on the assay readout?

Direct in vitro biochemical characterization (outside the yeast) of the nucleating properties of representative constructs of IAPP(IAPP variant)-Sup35(nucleation domain) in direct comparison with the properties of IAPP(IAPP variant) and the events in the context of the yeast cells could help to understand the molecular steps and why they can be used to report on amyloidogenicity of IAPP and its variants.

(1c) Some of the data is not consistent with previous findings of direct in vitro studies with IAPP variants. E.g. several studies have reported that the Ser20 to Gly T2D-related mutation enhances IAPP amyloidogenicity (e.g. Gao et al. JMB 2011). Also, Leu16 and Phe23 to Ala mutations were reported to decrease IAPP amyloidogenicity (Bakou et al. JBC 2017). I suggest that the authors directly compare their findings with previously published ones e.g. in form of a table in the Discussion part and comment on potential limitations of the screening approach.

(1d) How do the conditions of the cell assay account for differences between amyloid self-assembly kinetics and do end time points affect the results? Also, is nucleation ability related to the lag time of amyloid formation and the self- or cross-nucleating ability of IAPP/its variants alone or within the Sup35(nucleation domain)-containing construct?

2) The manuscript is too short, seems to have been hastily written, and does not help the reader -in particular non-specialists- to follow and compare the data analysis and conclusions also in the context of findings by others. For instance: (2a) Information helping non-experts to follow data analysis and conclusions is in many cases missing. This includes e.g. a more thorough description of figures in the figure legends (e.g. Fig. 1b, units of x axis? (logarithmic?), Fig. 1b. significance of the differences (and how this was calculated). Fig. 2c units missing, information on significance of differences/stats missing

Fig. 2c: legend missing. (2b) Important references/comparisons to results by others related to the findings (e.g. Westermark, Eisenberg, Radford, Raleigh, Kapurniotu, Gazit, Lansbury, Fraser groups) are missing in several cases or are incomplete (e.g. in "Results": "Our assay also captures the different amyloid propensity of IAPP animal variants that have previously been characterized...", in "Introduction" "Four of these identical amino acids are located in the previously described core of the fibrils formed in vitro by both proteins (NxGAI, positions 22, 24-26 in IAPP and 27, 29-31 in A β 42)."

3) Some of the claims should be toned down/rephrased and references to the work of others should be added: e.g. (a) "Besides comprehensively mapping for the first time the IAPP mutational landscape and identifying the likely structured core of the nucleating IAPP fibrils...". Large parts already done e.g. by some of the groups mentioned above and others. (b) "...suggesting a common mechanism of disruption of amyloid nucleation".. (e.g. papers on inhibitors of both Abeta and IAPP e.g. by Gazit, Kapurniotu, Hoyer, Eisenberg).

(Remarks on code availability)

Reviewer #3

(Remarks to the Author)

Badia and colleagues report their deep mutational scanning results on islet amyloid polypeptide (IAPP), an important protein for Type 2 diabetes. This paper is made much more interesting for a general scientific audience by the fact that this work follows a previous study from the same group looking at Amyloid beta, which like IAPP, forms potentially harmful aggregates. The data is convincing and the authors perform some interesting additional analyses to add context and show clinical relevance, including investigating phenotypes in UK BioBank. However, the figures are confusing at times and the paper suffers from a lack of key citations throughout the manuscript.

Major comments:

Despite reporting a deep mutational scan, there is no information about this technology whatsoever in the introduction or in the first section of the results where the study is described. Instead, the authors only mention their own previous work on Amyloid beta. I think this should be addressed by expanding the introduction with a paragraph about deep mutational scanning/multiplexed assays of variant effect and key citations, such as Fowler & Fields 2014 and Tabet et al. 2022. The authors should feel free to choose any of their favorite reviews or other relevant literature when writing this section. Nature Communications has a broad readership that may not already be familiar with this area, and citing landmark papers also contributes to discoverability of the paper.

The paper omits a number of key citations, and I hope that the editors will be flexible to allow the authors to include them all. Specifically, Fig. 1e summarizes results from "over ten previous studies" but these studies are not cited. Many PDB structures are referenced by their PDB accession, but the papers reporting these structures are not cited. When comparing Amyloid beta to IAPP, the authors use several aggregation and secondary structure predictors, as well as variant effect predictors. Shockingly, none of these tools are cited either despite most (or all) of them having associated peer-reviewed papers.

I found some of the figures to be quite confusing and perhaps a bit mixed up. I'm not sure why Fig. 1f and 1g are in the first figure, which otherwise handles non-DMS data, rather than part of Fig. 2. I found this abrupt shift in the type of data being presented very confusing. That said, I'm not sure that Fig. 1f adds very much, and I did not like the sideways version of it in Fig. 1g. I thought it added a lot of clutter, and I'm unsure why the authors didn't just color code the bar chart instead. I did really like the little cartoons at the top of Fig. 1g but I almost missed them because there was so much other stuff going on.

In Fig. 4a, 4b, and 4c, there are some full-length protein sequences listed to show what the variants correspond to. I found this added a lot of clutter and was quite confusing because it wasn't explained in the figure legend and the colored amino acids (that show the variant) are not super obvious, especially as someone who hasn't spent a lot of time looking at the sequence of IAPP. I think the authors should try to harmonize the various heatmaps as much as possible and also add clear titles to clarify things for readers. If the authors want to keep the full-length sequences as examples, maybe also increasing the font size of the inserted amino acids would help.

I don't think that Fig. 4d-g added very much and the plots are not particularly interesting. If the authors think it is important to retain them in the main text, it is essential that they are given titles in the figure panel. Currently they all look the same but are summarizing different types of mutations.

When referring to individual variants and individual positions, the authors use a highly abbreviated format (e.g. "S20G") that will make it difficult for advanced literature search methods in the future to resolve these variants. The authors should consider formatting variants using HGVS format, including the associated NP_ or ENSP accession number for IAPP. More

importantly, the authors do not actually specify a database accession number for the IAPP sequence used as the basis for their experiments. This should be included in the methods under the "Library design" section.

The authors have made their dataset available as an Excel table deposited in GEO, but the format is not very useful and it would be difficult for other researchers (or clinicians) to apply or reuse the data. The authors should make their data available using a more well-described format in a community repository like MaveDB, which is dedicated to hosting deep mutational scanning datasets like this one.

Minor comments:

In the introduction, paragraph 3, "one unique set-up" is mentioned but I thought this was very confusing. Are the authors referring to measuring multiple variants in a single assay?

Also in the introduction, paragraph 3, the authors say it is "currently unknown which mutations could increase IAPP aggregation rate", but that seems to have been written before they performed this study and should be updated.

At the end of the second paragraph in the results, the authors do not define "WT". This should probably be written out as "wild type".

In the first line of the second paragraph in the section "14 single amino acid deletions..." "multi-AA disrupt" should probably be "multi-AA deletions disrupt".

The authors refer to many insertion variants as "resulting from polymerase slippage" but I think they need to clarify if these are variants that were the result of slippage in their assay, or if they were designed as part of the library to simulate slippage that could happen in patients, leading to a clinical phenotype.

The authors use hierarchical clustering to separate IAPP into multiple segments. It was not obvious to me that this was the goal until I looked at the supplemental figure, so the authors should clarify this in the text.

To aid readers who are trying to interpret the data, the heatmaps like Fig. 2a and Fig. 2b should highlight the NNFGAIL segment that is mentioned repeatedly in the text along with the other existing annotations.

In the legend for Fig. 2b, the authors state "Variants that have been reported to not nucleate in the literature are indicated with a dot." Aren't all the variants in the figure indicated with the dot? This sentence suggests there are plotted variants that are not reported in the literature.

Fig. 2c has no figure legend.

The PDB structures in the Fig. 3c y-axis labels are inconsistently formatted with lowercase letters.

(Remarks on code availability)

The code is unlikely to be reusable or particularly useful for any other dataset, but it seems sufficient for a motivated future student to re-generate the figures.

Version 1:

Reviewer comments:

Reviewer #1

(Remarks to the Author)

The authors have done an excellent job addressing my comments, and I recommend the paper for publication in its current form.

(Remarks on code availability)

The authors have provided the relevant code and included a direct link as well.

Reviewer #2

(Remarks to the Author)

The authors addressed most of the comments sufficiently and the manuscript has been significantly improved. There are only a few remaining points to be addressed:

1) The supplementary tables were missing from the pdf file of the revised manuscript.

2) The authors claim "While this fusion peptide may not recapitulate all of the modifications that in vivo characterize biologically active IAPP, such as C-terminal amidation and formation of the disulfide bond between Cys 2 and Cys 7,". I think that the important information that the fusion proteins do not seem to recapitulate the cytotoxicity of IAPP oligomers or fibrils in the experimental setup used in the study should be included to the above sentence as well.

3) Comment 1b from my report: (a) The authors addressed this point in their response letter and concluded that Sup35N-

IAPP fusion is required to recruit endogenous Sup35 into aggregates. The corresponding Figure shown in their response letter should be also included in the supplementary part of the manuscript. (b) Direct evidence for the amyloid forming propensity e.g. by TEM of the aggregation nucleating agents i.e. the Sup35N-IAPP and selected Sup35N-IAPP mutant fusion proteins is still missing. I think that it is important to provide it to support the hypothesis that the output of their assay correlates to amyloid forming propensities.

(Remarks on code availability)

Reviewer #3

(Remarks to the Author)

I commend the authors for their diligent work addressing the wide spectrum of comments from the reviewers. I'm very happy with the revisions to the manuscript and enthusiastically support its publication. I have a few very minor comments below, including some optional additional suggestions regarding the figures.

Minor comments:

I don't think citation 69 is correct, as it cites the webpage for a collection of papers rather than an individual work. Presumably this should cite the current flagship paper (PMID 38057664) as per the gnomAD team's guidance.

Optional comments on the figures:

Figure 2a has the WT-like variants in white, but that means that the bars don't show up on the white background. The authors could address this by outlining the bars or changing the color of WT-like to gray.

There is a similar issue with Figure 4e, where the WT-like variants are colored white and not outlined with a box (since the box indicates a statistical significant difference). Again, perhaps gray boxes (or dark gray with white text) would work better here, although I understand the value of having the colors match the heat map color bar.

(Remarks on code availability)

The code and documentation appears sufficient to reproduce the figures and analysis in the paper.

The authors may wish to consider depositing the code in Zenodo or a similar archival service that provides a DOI.

We would like to thank the reviewers and the editor for the careful evaluation of our manuscript “Massively parallel quantification of mutational impact on IAPP amyloid formation” and for their valuable feedback. During the revision process, we took the opportunity to further strengthen our work with additional experimental work. More specifically, we have:

- Synthesized a new library which was cloned and selected under identical experimental conditions. This library was designed with the objective of generating a more complete map of IAPP single amino acid substitutions and insertions, using the same high-throughput setup applied to the rest of the data in the manuscript.
- Measured in vitro aggregation kinetics for synthetic IAPP variants by following Thioflavin-T fluorescence over time
- Measured the toxicity of IAPP variants featuring different nucleation scores
- Checked that the expression of un-fused IAPP is not conferring ability to grow in selective conditions
- Extracted total protein amounts for a subset of variants and confirmed that expression levels do not correlate with nucleation scores

All figures and corresponding text have been updated accordingly, and these revisions are highlighted throughout the revised manuscript text. Furthermore, we edited the discussion section incorporating the reviewers' suggestions.

Reviewer #1 (Remarks to the Author)

The manuscript by Badia, Battle, and Bolognesi focuses on the high-throughput quantification of over 1,600 genetic mutations and their effects on the amyloid aggregation of IAPP. Through their analysis, the authors identify critical residues, particularly the stretch from residue 15 to 32 as essential for the aggregation of the peptide. They also compare these results with previous data on the amyloid-beta peptide. The rationale behind this comparison is that the two peptides share significant sequence similarity. The authors find that the effects of mutations that slow down nucleation correlate between the two amyloids. However, when it comes to mutations that trigger or accelerate nucleation, a dataset from one amyloid cannot be used to predict mutational effects in the other.

This is a highly novel and interesting paper that makes important contributions to both method development and foundational science. The dataset they have obtained will be highly relevant and useful to the protein chemistry community. Overall, the manuscript is well written, and the findings are supported by the data. I believe this work will be of broad interest to the Nature Communications' readership. Thus, I am supportive of its publication, provided that the authors address my following minor concerns.

1) The authors employ a Sup35 fusion strategy in which IAPP sequences are fused to Sup35. Since Sup35 nucleation is required for yeast survival in the absence of adenine, variants that aggregate/ form amyloid fibrils in yeast are selectively enriched. The authors should clarify how they control for potential false negatives. There is a scenario in which a variant forms aggregates that are toxic to yeast, causing the cells to die or grow poorly despite successful nucleation. As a result, such variants may appear to have low nucleation potential, leading to

false negatives in the selection assay. Given the presence of WT control and the high number of mutants analysed, this is probably not a problem. However, I would encourage the authors to discuss this aspect in the manuscript.

We appreciate the reviewer's observation regarding the possibility of false negatives arising from aggregate-induced toxicity, which could reduce yeast growth and lead to misleadingly low nucleation scores. To address this, we compared the growth rates of yeast expressing Sup35-IAPP fusion constructs for nine variants spanning a range of nucleation scores. According to a one-way ANOVA, the growth rates of the tested variants are comparable, with no significant differences detected in either induced or uninduced conditions. This indicates that the constructs do not differentially affect cellular fitness independent of nucleation behavior. As in our previous work (Seuma, 2022 and Martín, 2025), these results support the conclusion that the selection assay reflects nucleation potential rather than general toxicity. The data and analysis are provided in Suppl. Fig. 1c, cited in the "Massively parallel quantification of IAPP amyloid nucleation" section of the Results and shown here:

Comparison of growth rates for IAPP variants with and without induced protein expression. Individually measured growth rates for selected IAPP variants classified as increasing (NS_inc), decreasing (NS_dec), or with WT-like nucleation (FDR = 0.1), assessed under non-inducing (no Cu⁺⁺) and inducing (Cu⁺⁺) expression conditions (n = 3 biological replicates per variant). One-way ANOVA detected no significant differences in growth among variants under either condition.

2) Similarly, the authors should clarify how they normalize for differences in protein expression levels across variants. For example, if a variant is expressed at lower levels, it might appear to have reduced nucleation potential, even if it can aggregate or form amyloids.

This is an important point. The current assay set-up does not allow for normalization for protein expression levels, so final protein concentration is a potential confounding factor. We have extracted total protein fractions from cells expressing WT IAPP as well as 7 IAPP variants (2 NS+, 1 WT-like and 4 NS-). While we find slight variation in expression levels, we do not find that variants with reduced nucleation scores are all significantly less expressed than WT, or that variants with higher nucleation scores are simply more highly expressed, suggesting that nucleation scores are an accurate reflection of the propensity of these sequences to initiate aggregation in the cell, rather than of their concentration. This Western blot is now included in Suppl. Fig. 1e and referred to in the “Massively parallel quantification of IAPP amyloid nucleation” section of the Results. Since this data only reflects the expression of a limited number of variants, we have also added a sentence in the discussion to acknowledge the limitations of the assay on this front.

Comparative expression of Sup35N-IAPP variants. Protein extracts from induced cultures were examined by western blotting using an anti-HA antibody to detect HA-tagged Sup35N-IAPP constructs. Variant labels are colour-coded based on nucleation score classification. An untagged lysate is shown as a negative control, and GAPDH is included as loading control.

3) Could the authors clarify how the analysis in Figure 1e was performed? Specifically, how were these variants selected? Were the aggregation conditions across these papers/published results comparable, or were the results normalized to a control, e.g., the WT IAPP?

We appreciate the reviewer’s concern. Heterogeneity in experimental conditions and reporting standards across the literature makes this challenging. To address this, we systematically collected all studies reporting Thioflavin T fluorescence measurements of IAPP variants relative to WT. Reaction conditions and the corresponding references, including DOIs for each study, are reported in column “Experimental conditions” of Suppl. Table 1.

Importantly, all variants were classified based on comparison to the WT measured within the same study, ensuring that each variant's behavior is interpreted relative to its own experimental control. Because most studies report only a few variants (typically 4–5), we opted for this comprehensive literature-based approach rather than arbitrarily selecting a single study. To improve clarity, we have also revised the Figure 1c legend (previous 1e) to explicitly indicate how variants are grouped as increasing, decreasing, or aggregating like WT, emphasizing that these classifications are always relative to the WT in each individual study.

4) The authors present a compelling analysis by mapping their nucleation scores onto high-resolution structures of amyloid fibrils formed by IAPP or amyloid-beta. By identifying key residues aligning with fibril cores, they provide support that the yeast-formed aggregates are amyloid-like and resemble those described in the literature. To further strengthen this claim, it would be helpful to include additional supporting evidence. For example, could the authors quantify IAPP-positive, SDS-insoluble fractions for a few relevant yeast variants (e.g., mutants within the fibril core and the WT) to assess whether there is any correlation with the nucleation score?

We appreciate the reviewer's suggestion. We agree that orthogonal validation is important, and in our view the degree of similarity between the known structural cores of IAPP fibrils and the mutational sensitivity patterns revealed by our atlas provides compelling evidence that the core of aggregates formed in yeast is similar to that of those amyloids for which high-resolution structures exist.

Regarding the proposed experiment, SDS-insolubility is an endpoint measurement that does not distinguish between amyloid fibrils and other aggregated or amorphous species; therefore, it would not provide structural resolution on whether yeast-formed aggregates adopt fibril-like conformations. Second, our assay reports on nucleation kinetics, not on the total amount of material accumulated at equilibrium. As such, endpoint fractionation is not an informative readout for the process we are measuring. We have indeed validated the kinetic nature of the scores directly. As described in response to points 5) and 6), *in vitro* ThT aggregation assays performed during revision show that the half-time of aggregation for synthetic IAPP peptides (not fused to Sup35N) scales with their nucleation scores for a representative subset of variants. This kinetic concordance provides the type of orthogonal support most relevant to the assay's interpretation.

5) The authors identify Cys7 to be an important residue for IAPP aggregation. However, it is not involved in a disulphide in amyloid structures from patients. What is the proposed role of this residue?

In our dataset, removal of either native cysteine (Cys2 or Cys7) consistently reduces nucleation, while additional cysteines introduced by substitution, insertion, or polymerase slippage, increase nucleation. This effect is unlikely to be explained by canonical disulphide bonding for several reasons:

- the yeast cytoplasm is a reducing environment (Morgan, 2012), and the SupN–IAPP constructs are not secreted through the ER where disulphides typically form;

- adjacent cysteines also boost nucleation in our assay, even though such pairs sterically could not establish a disulphide bond.

As the reviewer points out, a disulphide bond between these cysteines is also not observed in IAPP fibrils extracted from patients, nor in those formed in vitro. IAPP fibrils form in vitro regardless of disulphide status, and structural models of IAPP fibrils show that Cys2 and Cys7 are outside of the structured core (Rodríguez-Camargo, 2017). These points are noted in the Discussion section of the manuscript.

We also note that the effect of mutating cysteines or mutations to cysteines is striking in the first 9 residues of IAPP, but not in the rest of the sequence, nor in other amyloids previously studied employing the same assay (Seuma, 2022 and Martín, 2025).

During the revision process, we have measured ThT aggregation kinetics in reduced conditions for variants losing or acquiring cysteines in the N-terminal region (p.Cys7Thr and p.Cys7_Ala8insCysCys). We observe that the scaling between nucleation scores and lag time of the aggregation reaction is maintained if copper is available in the reaction buffer, while this is partially lost in the absence of copper. Copper substantially speeds up aggregation of p.Cys7_Ala8insCysCys, but not of p.Cys7Thr. This leads us to suggest that cysteines here may promote nucleation through a specific mechanism requiring multiple cysteines, for example through metal interactions and multimerization. The precise mechanism remains to be determined, and it is the current focus of a follow-up project in the lab.

Aggregation kinetics of IAPP variants in the presence and absence of Cu²⁺. Aggregation of IAPP WT, p.Cys7Thr, and p.Cys7_Asn8insCysCys peptides was monitored following Thioflavin T (ThT) fluorescence in the presence and the absence of 10 μ M Cu²⁺. Peptides were prepared at a final concentration of 5 μ M in 50 mM sodium phosphate buffer (pH 7.4) containing 20 μ M ThT, and fluorescence at 480 nm was measured every 10 minutes for 37 hours. Curves show mean fluorescence \pm SD of three technical replicates ($n = 3$). Both raw fluorescence values (top) and normalized fluorescence plots (bottom) are presented for comparison.

Correlation between aggregation half-time ($t_{1/2}$) in the presence and absence of Cu^{2+} and nucleation score for IAPP variants. $t_{1/2}$ values were obtained from normalized ThT fluorescence curves for IAPP WT, p.Cys7Thr, and p.Cys7_Asn8insCysCys peptides aggregated with or without 10 μM Cu^{2+} (curves shown above). Pearson correlation coefficients (r) and associated p -values are shown for both conditions. Vertical error bars represent the error of the nucleation score estimates.

6) Does the nucleation score reflect a thermodynamic property (e.g., stability of the final aggregated state), a kinetic parameter (e.g., rate of nucleation), or a combination of both?

We propose that nucleation scores likely primarily reflect kinetic effects on nucleation, not thermodynamic stability of the final aggregated state because of the following:

- 1) Agreement with the literature points to kinetics: variants with known kinetic effects on nucleation shift our score in the expected direction (as shown in Figure 1c in the manuscript). As most published studies do not report full kinetic traces or rate constants, this agreement is at the level of direction of effect, but it is consistently aligned with reaction rates rather than thermodynamic stability.
- 2) Independent evidence from other sequences in the same assay: for A β 42, ADan, and ABri (Seuma, 2021, Seuma, 2022 and Martín, 2025), we have shown that the same yeast-based readout correlates with nucleation rates measured in vitro by us and others.
- 3) Direct validation with synthetic peptides: to further strengthen our claim, during revision, we measured ThT fluorescence over time for three synthetic variants (data shown below). Changes in the nucleation scores scale with changes in the reaction half-time for these variants, again supporting a kinetic interpretation.
- 4) Lack of correlation with thermodynamic stability predictions: FoldX $\Delta\Delta\text{G}$ values (Schymkowitz, 2005) do not correlate with the nucleation scores, neither globally nor for specific variants, arguing against a thermodynamic stability interpretation (data shown below).

Together, these lines of evidence strongly indicate that the nucleation score captures kinetic modulation of amyloid nucleation, rather than the thermodynamic stability of the final fibril.

Aggregation kinetics of IAPP peptides. Aggregation of IAPP WT, p.Cys7Thr, and p.Ala13_Asn14insLeu peptides was monitored using a continuous Thioflavin T (ThT) fluorescence assay. Peptides were prepared at a final concentration of 5 μ M in 50 mM sodium phosphate buffer (pH 7.4) containing 20 μ M ThT, and fluorescence at 480 nm was recorded continuously for 20 hours. Each condition was measured in triplicate, and representative curves are shown. Both raw fluorescence values (top) and normalized fluorescence plots (bottom) are presented for comparison.

Correlation between aggregation half-time ($t_{1/2}$) and nucleation score for IAPP variants. $t_{1/2}$ values were derived from normalized ThT fluorescence curves for IAPP WT, p.Cys7Thr, and p.Ala13_Asn14insLeu peptides (curves shown above). Pearson correlation coefficients (r) and corresponding p -values are indicated. Vertical error bars represent the error of the nucleation score estimates, and horizontal error bars represent the standard deviation of the three independent measurements used to calculate $t_{1/2}$.

Correlation between FoldX-predicted $\Delta\Delta G$ and nucleation scores for IAPP substitutions. FoldX was used to calculate the change in stability ($\Delta\Delta G$, kcal/mol) for all IAPP substitutions across 8 available PDB structures. Pearson correlation coefficients (r) and associated p -values are indicated.

7) The authors should provide additional information on how the Nucleation Score was calculated in the section Nucleation scores and error estimates. For example, they should clarify what “log(F_i or wt input or output)” represents, to make this clearer to a wider readership. I understand that this refers to the in vivo enrichment and that more details were provided in the original ELife 2021 paper. However, it would be helpful to include more information here to make the manuscript more self-contained.

We thank the reviewer for this helpful comment. We have revised the Nucleation scores and error estimates section to provide a clearer explanation of how the Nucleation Score is calculated, including the meaning of “log(F_i or WT input or output)”.

(Remarks on code availability)

The link provided directs you to the GitHub page of the team. I would recommend that the authors report the link that sends directly to the project's folder. All scripts to make all the figures and tables are present. Instructions on how to install and run the code are also present. Scripts are commented to help the analysis.

We thank the reviewer for pointing this detail out. We have included the direct link to the specific project folder within our GitHub repository in the Data Availability section of the Methods.

Reviewer #2 (Remarks to the Author):

The manuscript presents data derived from deep mutational scanning of the 37-residue polypeptide IAPP, a highly amyloidogenic polypeptide whose amyloid self-assembly is closely associated to T2D pathogenesis. The aim was to identify mutations or other sequence changes that affect the amyloidogenic potential of IAPP by deep mutational scanning. The authors used MAVEs and deep indel mutagenesis methodology (DIM) recently used to study Abeta42 of AD (ref 31). They generated 1663 IAPP variants which comprised variants with single amino acid substitutions, insertions or deletions, and variants with truncations, larger internal deletions, and sequence alterations which could result from polymerase slippage. Amyloidogenic potentials (termed “nucleation score” or NS in comparison to (human) IAPP) were quantified by using the yeast cell assay (ref. 53) which was also used in the Abeta42 studies (ref 53) and which they adapted for studying IAPP.

Overall the results are very impressive and consistent with results by others based on structural models of IAPP fibrils and early aggregates and derived from various structure-amyloidogenicity studies which were done using classical biochemical/biophysical methodology. In summary, the data is consistent with IAPP (15-32) being a core sequence of IAPP fibrils and early aggregates. Also, it is shown that mutations have a more drastic effect in IAPP (21-27) or NNFGAIL which was earlier found to be a key amyloid core segment of IAPP. The approach could find broad applicability in the amyloid field and beyond.

Additional comments

1) The results are derived from a yeast cell assay that is based on the capability of a fusion construct of IAPP with the nucleation domain of Sup35 to nucleate Sup35 amyloid formation which is required for yeast survival under the lack of adenine. The latter one provides a functional readout (NS or nucleation score) for the Sup35 amyloid nucleating ability of IAPP or its diverse variants, which should according to the authors' hypothesis correlate with the amyloidogenic potential of IAPP and its variants as found in the case of analogous Abeta42/Sup35 constructs (ref. 31).

(1a) Regarding the proof-of-principle for the suitability of the cell assay: In Fig. 1b the authors present data related to the suitability of their assay for quantifying amyloidogenic potentials. However, it is unclear whether the differences between the relative growth rates of the different IAPP variants are significant because information about the x-axis (relative growth units and scale (is it a logarithmic scale?)) and the significance of the differences between the data of IAPP and its variants is missing.

We thank the reviewer for noting the need to clarify the x-axis. In Fig. 1d (Fig. 1b in the previous version of the manuscript), the x-axis represents relative growth (%), calculated as the ratio of colonies growing on selective medium to those on non-selective medium, multiplied by 100 and shown on a linear scale. We have updated the x-axis label to reflect this. Statistical differences between human IAPP and the animal variants were evaluated using the Wilcoxon

rank-sum test with Benjamini–Hochberg correction, and significant differences are now indicated in the figure.

(1b) The authors' conclusions are based exclusively on the readout of their yeast cell assay which is an indirect assay. However, I am missing direct biochemical assay validation experiments in the manuscript and there are some important questions on the principle and the applicability of the assay. For instance, which is the nucleating agent? Is this amyloid-like fibrillary assemblies formed by the construct of IAPP (or its amyloidogenic IAPP variant) with Sup35 (nucleation domain). Are Sup35 amyloid-like fibrils cross-nucleated by the fibrils of the construct of IAPP (amyloidogenic IAPP) /Sup35(nucleation domain)?

Sup35 nucleation is not detected under overexpression of the Sup35 nucleation domain alone. This was originally reported by Chandramowliswaran et al. (Chandramowliswaran, 2019) and confirmed by us (Seuma, 2021 and Seuma, 2022), fusion of amyloidogenic sequences (A β (1-42), PrP (90-230), α -synuclein (61-95) to the Sup35 nucleation domain is required for recruitment of endogenous Sup35 into aggregates, allowing growth on –Ade. In addition, as shown Chandramowliswaran et al. also show that no growth on –Ade is observed when amyloidogenic sequences (A β 1-42, PrP(90-230)) and the Sup35 nucleation domain are co-expressed in trans.

To address these specific concerns, we have now addressed whether this also holds true for IAPP. The assay here below shows how growth in the lack of adenine is observed only when Sup35 is fused to the IAPP sequence. Sup35 overexpression alone, or co-expression of IAPP and Sup35 in trans, does not trigger growth in the lack of adenine (see image below). This suggests that the Sup35N-IAPP fusion is required to recruit endogenous Sup35 into aggregates.

Overexpression of Sup35N and IAPP allows growth in –Ade in cis, but not in trans.

Yeast cells expressing different IAPP constructs were grown under selective (–Ade) and non-selective conditions. Protein expression was induced for 24 hours with 100 μM Cu^{2+} and 2% galactose. Growth is observed only when IAPP is fused to Sup35N, whereas co-overexpression of Sup35N and IAPP in trans does not support growth on –Ade, showing that nucleation of endogenous Sup35 requires the overexpression of the fusion construct.

What is the effect of kinetics of amyloid-like fibril formation and fibril thermodynamic stability of IAPP (IAPP variants) and of their constructs with Sup35 (nucleation domain) on the assay output?

We propose that nucleation scores likely primarily reflect kinetic effects on nucleation, not thermodynamic stability of the final aggregated state because of the following:

- 1) Agreement with the literature points to kinetics: variants with known kinetic effects on nucleation shift our score in the expected direction (as shown in Figure 1c in the manuscript). As most published studies do not report full kinetic traces or rate constants, this agreement is at the level of direction of effect, but it is consistently aligned with reaction rates rather than thermodynamic stability.
- 2) Independent evidence from other sequences in the same assay: for A β 42, ADan, and ABri (Seuma, 2021, Seuma, 2022 and Martín, 2025), we have shown that the same yeast-based readout correlates with nucleation rates measured in vitro by us and others.
- 3) Direct validation with synthetic peptides: to further strengthen our claim, during revision, we measured ThT fluorescence over time for three synthetic variants (data shown below). Changes in the nucleation scores scale with changes in the reaction half-time for these variants, again supporting a kinetic interpretation.
- 4) Lack of correlation with thermodynamic stability predictions: FoldX $\Delta\Delta\text{G}$ values (Schymkowitz, 2005) do not correlate with the nucleation scores, neither globally nor for specific variants, arguing against a thermodynamic stability interpretation (data shown below).

Together, these lines of evidence strongly indicate that the nucleation score captures kinetic modulation of amyloid nucleation, rather than the thermodynamic stability of the final fibril.

Aggregation kinetics of IAPP peptides. Aggregation of IAPP WT, p.Cys7Thr, and p.Ala13_Asn14insLeu peptides was monitored using a continuous Thioflavin T (ThT) fluorescence assay. Peptides were prepared at a final concentration of 5 μ M in 50 mM sodium phosphate buffer (pH 7.4) containing 20 μ M ThT, and fluorescence at 480 nm was recorded continuously for 20 hours. Each condition was measured in triplicate, and representative curves are shown. Both raw fluorescence values (top) and normalized fluorescence plots (bottom) are presented for comparison.

Correlation between aggregation half-time ($t_{1/2}$) and nucleation score for IAPP variants. $t_{1/2}$ values were derived from normalized ThT fluorescence curves for IAPP WT, p.Cys7Thr, and p.Ala13_Asn14insLeu peptides (curves shown above). Pearson correlation coefficients (r) and corresponding p -values are indicated. Vertical error bars represent the error of the nucleation score estimates, and horizontal error bars represent the standard deviation of the three independent measurements used to calculate $t_{1/2}$.

Correlation between FoldX-predicted $\Delta\Delta G$ and nucleation scores for IAPP substitutions. FoldX was used to calculate the change in stability ($\Delta\Delta G$, kcal/mol) for all IAPP substitutions across 8 available PDB structures^{14-17,19}. Pearson correlation coefficients (r) and associated p-values are indicated.

What is the effect of potential intrinsic cytotoxicity of IAPP-related aggregates and fibrils on the assay readout?

We appreciate the reviewer's observation regarding the possibility of false negatives arising from aggregate-induced toxicity, which could reduce yeast growth and lead to misleadingly low nucleation scores. To address this, we compared the growth rates of yeast expressing Sup35-IAPP fusion constructs for nine variants spanning a range of nucleation scores. According to a one-way ANOVA, the growth rates of the tested variants are comparable, with no significant differences detected in either induced or uninduced conditions. This indicates that the constructs do not differentially affect cellular fitness independent of nucleation behavior. As in our previous work (Seuma, 2022 and Martín, 2025), these results support the conclusion that the selection assay reflects nucleation potential rather than general toxicity. The data and analysis are provided in Suppl. Fig. 1c, cited in the "Massively parallel quantification of IAPP amyloid nucleation" section of the Results and shown here below:

Direct in vitro biochemical characterization (outside the yeast) of the nucleating properties of representative constructs of IAPP(IAPP variant)-Sup35(nucleation domain) in direct comparison with the properties of IAPP(IAPP variant) and the events in the context of the yeast cells could help to understand the molecular steps and why they can be used to report on amyloidogenicity of IAPP and its variants.

In order to better relate nucleation scores to biophysical parameters, we measured ThT fluorescence over time for three synthetic variants (data shown below). Changes in the nucleation scores scale with changes in the reaction half-time for these variants, thus supporting a kinetic interpretation.

Correlation between aggregation half-time ($t_{1/2}$) and nucleation score for IAPP variants.

$t_{1/2}$ values were derived from normalized ThT fluorescence curves for IAPP WT, p.Cys7Thr, and p.Ala13_Asn14insLeu peptides (curves shown above). Pearson correlation coefficients (r) and corresponding p -values are indicated. Vertical error bars represent the error of the nucleation score estimates, and horizontal error bars represent the standard deviation of the three independent measurements used to calculate $t_{1/2}$.

(1c) Some of the data is not consistent with previous findings of direct in vitro studies with IAPP variants. E.g. several studies have reported that the Ser20 to Gly T2D-related mutation enhances IAPP amyloidogenicity (e.g. Gao et al. JMB 2011). Also, Leu16 and Phe23 to Ala mutations were reported to decrease IAPP amyloidogenicity (Bakou et al. JBC 2017). I suggest that the authors directly compare their findings with previously published ones e.g. in form of a table in the Discussion part and comment on potential limitations of the screening approach.

The manuscript contains a comparison of nucleation scores obtained in this study with previously characterized IAPP variants, which now has been extended and includes, among others, those cited by the reviewer. This is summarized in Figure 1e and Supplementary Table S1, where any discrepancies between nucleation scores and estimates from ThT fluorescence traces are highlighted in gray.

To generate a more complete map of IAPP single amino acid substitutions and insertions and to increase discrimination power also for those variants close to WT, during the revision process, we synthesized a new library which was cloned and selected under identical experimental conditions. Details on the library design, cloning strategy, transformation efficiencies, and DNA sequencing data are provided in the *Methods* section, together with the methods used to merge the datasets and calibrate the nucleation scores across libraries for direct comparison. Notably, with these additional replicates, nucleation scores for the Ser20Gly, p.Leu16Ala, and p.Phe23Ala variants are significantly different from WT and

consistent with previously reported findings: a moderate increase for p.Ser20Gly (Xu, 2022, Akter, 2016, Cao, 2012 & Young, 2017) and a moderate decrease for p.Leu16Ala, and p.Phe23Ala (Bakou, 2017).

While some early Asian studies reported enrichment of p.Ser20Gly in T2D (e.g. Seino et al, 2001), multiple other studies from Japan, China, and Korea did not replicate this finding (summarized in Westermark, 2011), and the variant does not appear in large genome-wide datasets as a confirmed T2D locus (Lam, 2013). Given the lack of replication in modern large cohorts and different populations, we prefer to avoid confidently calling p.Ser20Gly T2D-associated.

(1d) How do the conditions of the the cell assay account for differences between amyloid self-assembly kinetics and do end time points affect the results? Also, is nucleation ability related to the lag time of amyloid formation and the self- or cross-nucleating ability of IAPP/its variants alone or within the Sup35(nucleation domain)-containing construct?

The experiments and response to 1b address both these points.

2) The manuscript is too short, seems to have been hastily written, and does not help the reader -in particular non-specialists- to follow and compare the data analysis and conclusions also in the context of findings by others. For instance:

(2a) Information helping non-experts to follow data analysis and conclusions is in many cases missing. This includes e.g. a more thorough description of figures in the figure legends (e.g. Fig. 1b, units of x axis? (logarithmic?), Fig. 1b. significance of the differences (and how this was calculated). Fig. 2c units missing, information on significance of differences/stats missing Fig. 2c: legend missing.

We have clarified figure legends throughout the manuscript and added additional details in the Methods section. We have now included a legend for panel 2e (previously 2c). Additionally, 1b is now 1d, and the new 1b includes the testing. All legends now provide relevant details, including units, whether axes are logarithmic, and the significance of differences along with the statistical tests used.

(2b) Important references/comparisons to results by others related to the findings (e.g. Westermark, Eisenberg, Radford, Raleigh, Kapurniotu, Gazit, Lansbury, Fraser groups) are missing in several cases or are incomplete (e.g. in "Results": "Our assay also captures the different amyloid propensity of IAPP animal variants that have previously been characterized....", in "Introduction" "Four of these identical amino acids are located in the previously described core of the fibrils formed in vitro by both proteins (NxGAI, positions 22, 24-26 in IAPP and 27, 29-31 in A β 42)."

We thank the reviewer for this helpful suggestion. In revising the manuscript, we have updated the relevant sections to include all appropriate references. Some of the studies suggested by the reviewer were not previously cited and have now been incorporated, while other references that were already cited elsewhere in the manuscript have been added to the specific points highlighted by the reviewer for completeness.

Animal variant studies (added to Results where previously missing):

- Ridgway, T. *Biophys. J.* 118, 1142–1151 (2020)
- Fortin, J. S. & Benoit-Biancamano, M.-O. *Amyloid* 22, 194–202 (2015)
- Akter, R. et al. *Isr. J. Chem.* 57, 750–761 (2017)

Core amyloidogenic segments of IAPP and A β 42 (added to Introduction where previously missing):

- Gilead, S. & Gazit, E. *Exp Diabetes Res.* 2008, 256954
- Soriaga, A. B. et al. *J. Phys. Chem. B* 120, 5810–5816 (2016)
- Abedini, A. & Raleigh, D. P. *J. Mol. Biol.* 355, 274–281 (2006)
- Krotee, P. et al. *eLife* 6:e19273 (2017)
- Röder, S. et al., *Nat. Struct. Mol. Biol.* 2020

These updates ensure that statements on IAPP amyloid propensity and core fibril residues are now fully supported by the literature.

3) Some of the claims should be toned down/rephrased and references to the work of others should be added: e.g. (a) “Besides comprehensively mapping for the first time the IAPP mutational landscape and identifying the likely structured core of the nucleating IAPP fibrils...”. Large parts already done e.g. by some of the groups mentioned above and others. (b) “..suggesting a common mechanism of disruption of amyloid nucleation”.. (e.g. papers on inhibitors of both Abeta and IAPP e.g. by Gazit, Kapurniotu, Hoyer, Eisenberg).

We have rephrased the first statement to acknowledge previous work as follows:

“Besides comprehensively mapping for the first time the IAPP mutational landscape, we identify a likely structured core of the nucleating IAPP fibrils that is consistent with and adds granularity to what has been suggested by previous studies.”

When it comes to the comment on mutations decreasing nucleation, we don’t want to suggest that the mechanisms of amyloid inhibition by small molecules are or can necessarily be the same as the mechanism by which mutations slow down the process. It may be, but our paper does not provide data in any of these directions, and we’d prefer to avoid suggesting this. We have, however, included a relevant reference to peptide inhibitors :

- Tatarek-Nossol et al. *Chem Biol* (2005)

Reviewer #3 (Remarks to the Author):

Badia and colleagues report their deep mutational scanning results on islet amyloid polypeptide (IAPP), an important protein for Type 2 diabetes. This paper is made much more interesting for a general scientific audience by the fact that this work follows a previous study from the same group looking at Amyloid beta, which like IAPP, forms potentially harmful aggregates. The data is convincing and the authors perform some interesting additional

analyses to add context and show clinical relevance, including investigating phenotypes in UK BioBank. However, the figures are confusing at times and the paper suffers from a lack of key citations throughout the manuscript.

Major comments:

Despite reporting a deep mutational scan, there is no information about this technology whatsoever in the introduction or in the first section of the results where the study is described. Instead, the authors only mention their own previous work on Amyloid beta. I think this should be addressed by expanding the introduction with a paragraph about deep mutational scanning/multiplexed assays of variant effect and key citations, such as Fowler & Fields 2014 and Tabet et al. 2022. The authors should feel free to choose any of their favorite reviews or other relevant literature when writing this section. Nature Communications has a broad readership that may not already be familiar with this area, and citing landmark papers also contributes to discoverability of the paper.

We thank the reviewer for this helpful suggestion. We have added a paragraph in the Introduction that references multiplexed assays of variant effects (MAVEs), highlighting their general principle, applications, and key references, to provide context for readers who may be less familiar with this approach.

The paper omits a number of key citations, and I hope that the editors will be flexible to allow the authors to include them all. Specifically, Fig. 1e summarizes results from "over ten previous studies" but these studies are not cited. Many PDB structures are referenced by their PDB accession, but the papers reporting these structures are not cited. When comparing Amyloid beta to IAPP, the authors use several aggregation and secondary structure predictors, as well as variant effect predictors. Shockingly, none of these tools are cited either despite most (or all) of them having associated peer-reviewed papers.

All corresponding citations have been included in the revised version of the manuscript.

I found some of the figures to be quite confusing and perhaps a bit mixed up. I'm not sure why Fig. 1f and 1g are in the first figure, which otherwise handles non-DMS data, rather than part of Fig. 2. I found this abrupt shift in the type of data being presented very confusing. That said, I'm not sure that Fig. 1f adds very much, and I did not like the sideways version of it in Fig. 1g. I thought it added a lot of clutter, and I'm unsure why the authors didn't just color code the bar chart instead. I did really like the little cartoons at the top of Fig. 1g but I almost missed them because there was so much other stuff going on.

We thank the reviewer for pointing out this aspect of the manuscript. In response, we have reorganized the figure panels to improve clarity. Figure 1 now focuses solely on showing the controls of our experimental approach and comparisons with previous literature. In parallel, we have also color-coded the histograms according to FDR categories and removed the frequency plots, as suggested, and organized them in a separate figure (Figure 2a in the revised version of the manuscript).

In Fig. 4a, 4b, and 4c, there are some full-length protein sequences listed to show what the variants correspond to. I found this added a lot of clutter and was quite confusing because it wasn't explained in the figure legend and the colored amino acids (that show the variant) are not super obvious, especially as someone who hasn't spent a lot of time looking at the sequence of IAPP. I think the authors should try to harmonize the various heatmaps as much as possible and also add clear titles to clarify things for readers. If the authors want to keep the full-length sequences as examples, maybe also increasing the font size of the inserted amino acids would help.

We appreciate the reviewer's suggestion. We have removed the full-length sequences from all panels and omitted amino-acid color highlighting. We also increased the title font sizes and rearranged the figure panels to improve clarity and readability. We believe these revisions address the reviewer's concerns and result in a cleaner and more accessible figure.

I don't think that Fig. 4d-g added very much and the plots are not particularly interesting. If the authors think it is important to retain them in the main text, it is essential that they are given titles in the figure panel. Currently they all look the same but are summarizing different types of mutations.

Following the reviewer's suggestion, we removed the plots that did not add substantial interpretative value and retained only the panels illustrating the regional effects of single-amino-acid insertions, which we consider important for presenting the results. We also added titles to facilitate interpretation of the figure.

When referring to individual variants and individual positions, the authors use a highly abbreviated format (e.g. "S20G") that will make it difficult for advanced literature search methods in the future to resolve these variants. The authors should consider formatting variants using HGVS format, including the associated NP_ or ENSP accession number for IAPP. More importantly, the authors do not actually specify a database accession number for the IAPP sequence used as the basis for their experiments. This should be included in the methods under the "Library design" section.

We thank the reviewer for this valuable suggestion aimed at improving the accessibility of our data. In response, we have reformatted the variants using HGVS nomenclature and included the appropriate NP_ accession number for the IAPP sequence in the Methods section under "Library design." We also provide, in Supplementary Table S2, a column with HGVS nomenclature referenced to preproIAPP and an additional column referenced to the mature peptide.

The authors have made their dataset available as an Excel table deposited in GEO, but the format is not very useful and it would be difficult for other researchers (or clinicians) to apply or reuse the data. The authors should make their data available using a more well-described format in a community repository like MaveDB, which is dedicated to hosting deep mutational scanning datasets like this one.

In response to the reviewer's comment, we have now deposited the full dataset in MaveDB. The dataset is available under the accession urn:mavedb:00001253-a. We have also added

a corresponding statement in the Data Availability subsection of the Methods section of the revised manuscript to reflect this addition.

Minor comments:

In the introduction, paragraph 3, "one unique set-up" is mentioned but I thought this was very confusing. Are the authors referring to measuring multiple variants in a single assay?

We agree that this should be clearer. Aggregation measurements available in the literature are the result of experiments run in a quite wide range of conditions, making comparative analysis more challenging, with less than a handful of variants characterized in the same set-up. We now explicitly phrase this as:

"The challenges of quantifying amyloid nucleation rates for multiple variants in the same conditions has so far partially limited our understanding of the sequence-nucleation relationship for IAPP".

Also in the introduction, paragraph 3, the authors say it is "currently unknown which mutations could increase IAPP aggregation rate", but that seems to have been written before they performed this study and should be updated.

We have revised this sentence in the current version of the manuscript, which now reads:

"Only a few IAPP variants have been characterized, and the vast majority have been shown to decrease IAPP nucleation, making it difficult to predict whether novel population variants might increase the risk of developing T2D."

At the end of the second paragraph in the results, the authors do not define "WT". This should probably be written out as "wild type".

This correction has been implemented in the revised version of the manuscript.

In the first line of the second paragraph in the section "14 single amino acid deletions..." "multi-AA disrupt" should probably be "multi-AA deletions disrupt".

We have corrected this in the revised version of the manuscript.

The authors refer to many insertion variants as "resulting from polymerase slippage" but I think they need to clarify if these are variants that were the result of slippage in their assay, or if they were designed as part of the library to simulate slippage that could happen in patients, leading to a clinical phenotype.

We have clarified this point in the Results section by replacing the phrase "resulting from polymerase slippage" with "designed sequence variants simulating polymerase slippage (i.e., replication errors that create small insertions or deletions due to transient misalignment of the polymerase with the template)." This revision specifies that these variants were intentionally designed as part of the library and were not generated spontaneously during library construction or in any subsequent experimental steps.

The authors use hierarchical clustering to separate IAPP into multiple segments. It was not obvious to me that this was the goal until I looked at the supplemental figure, so the authors should clarify this in the text.

We thank the reviewer for this comment, which prompted us to reconsider our approach. We realized that hierarchical clustering was not essential for defining segments and could potentially add confusion. For clarity and simplicity, we have omitted clustering and now define regions based on the percentage of mutations that increase or decrease nucleation, as determined from the FDR-based maps (Supplementary Figure 6b). The criteria used to define these regions are also described in the Data analysis subsection of the Methods section of the manuscript.

To aid readers who are trying to interpret the data, the heatmaps like Fig. 2a and Fig. 2b should highlight the NNFGAIL segment that is mentioned repeatedly in the text along with the other existing annotations.

We have highlighted this segment in Figure 2 and Supplementary Figure 3 in the revised version of the manuscript.

In the legend for Fig. 2b, the authors state "Variants that have been reported to not nucleate in the literature are indicated with a dot." Aren't all the variants in the figure indicated with the dot? This sentence suggests there are plotted variants that are not reported in the literature.

We thank the reviewer for pointing out this ambiguity. Our original sentence referred to a dot symbol placed next to the IAPP animal sequences written below to the plot, not to the dots representing individual data points in the boxplots. To prevent confusion, we have replaced this dot with a cross (×) and clarified the figure legend.

Fig. 2c has no figure legend.

We have corrected Figure 2 legend in the revised version of the manuscript.

The PDB structures in the Fig. 3c y-axis labels are inconsistently formatted with lowercase letters.

We have corrected this in the revised version of the manuscript.

Reviewer #3 (Remarks on code availability):

The code is unlikely to be reusable or particularly useful for any other dataset, but it seems sufficient for a motivated future student to re-generate the figures.

Our GitHub folder (link) includes comments to help installing and running the code which we hope should guide anyone to replicate our analysis.

References

Akter, R. et al. Evolutionary Adaptation and Amyloid Formation: Does the Reduced Amyloidogenicity and Cytotoxicity of Ursine Amylin Contribute to the Metabolic Adaption of Bears and Polar Bears? *Isr. J. Chem.* 57, 750–761 (2017).

Bakou, M., Hille, K., Kracklauer, M., Spanopoulou, A., Frost, C. V., Malideli, E., Yan, L.-M., Caporale, A., Zacharias, M. & Kapurniotu, A. Key aromatic/hydrophobic amino acids controlling a cross-amyloid peptide interaction versus amyloid self-assembly. *J. Biol. Chem.* 292, 14587–14602 (2017).

Cao, P., Tu, L.-H., Abedini, A., Levsh, O., Akter, R., Patsalo, V., Schmidt, A. M. & Raleigh, D. P. Sensitivity of amyloid formation by human islet amyloid polypeptide to mutations at residue 20. *J. Mol. Biol.* 421, 282–295 (2011).

Chandramowliswaran, P. et al. Mammalian amyloidogenic proteins promote prion nucleation in yeast. *J. Biol. Chem.* 293, 3436–3450 (2018).

Lam, V. K. L. et al. Genetic associations of type 2 diabetes with islet amyloid polypeptide processing and degrading pathways in Asian populations. *PLoS One* 8, e62378 (2013).

Martín, M. & Bolognesi, B. Massive mutagenesis reveals an incomplete amyloid motif in Bri2 that turns amyloidogenic upon C-terminal extension. *Proc. Natl. Acad. Sci. U.S.A.* 122, e2415521122 (2025).

Morgan, B., Ezeriņa, D., Amoako, T. N. E., Riemer, J., Seedorf, M. & Dick, T. P. Multiple glutathione disulfide removal pathways mediate cytosolic redox homeostasis. *Nat. Chem. Biol.* 9, 119–125 (2012).

Rodríguez Camargo, D. C. et al. The redox environment triggers conformational changes and aggregation of hIAPP in Type II Diabetes. *Sci. Rep.* 7, 44041 (2017).

Schymkowitz, J., Borg, J., Stricher, F., Nys, R., Rousseau, F. & Serrano, L. The FoldX web server: an online force field. *Nucleic Acids Res.* 33, W382–W388 (2005).

Seino, S. S20G mutation of the amylin gene is associated with Type II diabetes in Japanese. *Diabetologia* 44, 906–909 (2001).

Seuma, M., Faure, A., Badia, M., Lehner, B. & Bolognesi, B. The genetic landscape for amyloid beta fibril nucleation accurately discriminates familial Alzheimer's disease mutations. *Elife* 10, e63364 (2021).

Seuma, M., Lehner, B. & Bolognesi, B. An atlas of amyloid aggregation: the impact of substitutions, insertions, deletions and truncations on amyloid beta fibril nucleation. *Nat. Commun.* 13, 7084 (2022).

Westermarck, P., Andersson, A. & Westermarck, G. T. Islet amyloid polypeptide, islet amyloid, and diabetes mellitus. *Physiol. Rev.* 91, 795–826 (2011).

Xu, Y. et al. Tuning the rate of aggregation of hIAPP into amyloid using small-molecule modulators of assembly. *Nat. Commun.* 13, 1040 (2022).

Young, L. M., Tu, L.-H., Raleigh, D. P., Ashcroft, A. E. & Radford, S. E. Understanding copolymerization in amyloid formation by direct observation of mixed oligomers. *Chem. Sci.* 8, 503–509 (2017).

We would like to thank the reviewers and the editor for the careful evaluation of our manuscript “Massively parallel quantification of mutational impact on IAPP amyloid formation” and for their valuable feedback. All figures and corresponding text have been updated accordingly incorporating the reviewers' suggestions.

Reviewer #1 (Remarks to the Author):

The authors have done an excellent job addressing my comments, and I recommend the paper for publication in its current form.

Reviewer #1 (Remarks on code availability):

The authors have provided the relevant code and included a direct link as well.

We thank Reviewer #1 for the positive assessment and for recommending the manuscript for publication.

Reviewer #2 (Remarks to the Author):

The authors addressed most of the comments sufficiently and the manuscript has been significantly improved. There are only a few remaining points to be addressed:

We thank Reviewer #2 for the positive assessment and for all the suggestions.

1) The supplementary tables were missing from the pdf file of the revised manuscript.

We are now providing all Supplementary Tables as Supplementary Data due to their size.

2) The authors claim “While this fusion peptide may not recapitulate all of the modifications that in vivo characterize biologically active IAPP, such as C-terminal amidation and formation of the disulfide bond between Cys 2 and Cys 7,”. I think that the important information that the fusion proteins do not seem to recapitulate the cytotoxicity of IAPP oligomers or fibrils in the experimental setup used in the study should be included to the above sentence as well.

We agree that cytotoxicity is an important aspect of islet amyloid polypeptide biology. However, it is beyond the scope of this work, which focuses exclusively on amyloid nucleation. Notably, the lack of toxicity of islet amyloid polypeptide variants to yeast cells is a deliberate advantage, as it prevents confounding effects on the selection and enables to employ yeast at scale to obtain a clean readout of nucleation, as noted in the Results.

3) Comment 1b from my report: (a) The authors addressed this point in their response letter and concluded that Sup35N-IAPP fusion is required to recruit endogenous Sup35 into aggregates. The corresponding Figure shown in their response letter should be also included

in the supplementary part of the manuscript. (b) Direct evidence for the amyloid forming propensity e.g. by TEM of the aggregation nucleating agents i.e. the Sup35N-IAPP and selected Sup35N-IAPP mutant fusion proteins is still missing. I think that it is important to provide it to support the hypothesis that the output of their assay correlates to amyloid forming propensities.

a) We have included this panel as Supplementary Figure 1a, together with the corresponding figure legend, and added the experimental procedure to the Methods section under the heading *"Growth assays with fused and unfused Sup35N-IAPP constructs."*

b) We agree that direct structural evidence would further strengthen the link between assay output and amyloid-forming propensity. Structural characterization of Sup35N amyloid fusions is being carried out in parallel in our laboratory. Since this work is part of an ongoing, dedicated structural study, we consider it beyond the scope of the present manuscript.

Reviewer #3 (Remarks to the Author):

I commend the authors for their diligent work addressing the wide spectrum of comments from the reviewers. I'm very happy with the revisions to the manuscript and enthusiastically support its publication. I have a few very minor comments below, including some optional additional suggestions regarding the figures.

We thank Reviewer #3 for the careful evaluation of the revised manuscript, the positive assessment of the changes made, and the continued support for publication.

Minor comments:

I don't think citation 69 is correct, as it cites the webpage for a collection of papers rather than an individual work. Presumably this should cite the current flagship paper (PMID 38057664) as per the gnomAD team's guidance.

We updated the citation 69 (now reference 70) to the current flagship paper.

Optional comments on the figures:

Figure 2a has the WT-like variants in white, but that means that the bars don't show up on the white background. The authors could address this by outlining the bars or changing the color of WT-like to gray.

There is a similar issue with Figure 4e, where the WT-like variants are colored white and not outlined with a box (since the box indicates a statistical significant difference). Again, perhaps

gray boxes (or dark gray with white text) would work better here, although I understand the value of having the colors match the heat map color bar.

We incorporated all the aesthetic suggestions in Figure 2a and 4e, and in Supplementary Figure 7.

Reviewer #3 (Remarks on code availability):

The code and documentation appears sufficient to reproduce the figures and analysis in the paper.

The authors may wish to consider depositing the code in Zenodo or a similar archival service that provides a DOI.

We have also deposited all the code for the analysis into Zenodo.